# Classically Approximating Variational Quantum Machine Learning with Random Fourier Features

**Jonas Landman**
University of Edinburgh
QC Ware

**Slimane Thabet**
LIP6, Sorbonne Université
PASQAL SAS

**Constantin Dalyac**
LIP6, Sorbonne Université
PASQAL SAS

**Hela Mhiri**
LIP6, Sorbonne Université
ENSTA Paris

**Elham Kashefi**
University of Edinburgh
LIP6, Sorbonne Université

## Abstract

Many applications of quantum computing in the near term rely on variational quantum circuits (VQCs). They have been showcased as a promising model for reaching a quantum advantage in machine learning with current noisy intermediate scale quantum computers (NISQ). It is often believed that the power of VQCs relies on their exponentially large feature space, and extensive works have explored the expressiveness and trainability of VQCs in that regard. In our work, we propose a classical sampling method that can closely approximate most VQCs with Hamiltonian encoding, given only the description of their architecture. It uses the seminal proposal of Random Fourier Features (RFF) and the fact that VQCs can be seen as large Fourier series. We show theoretically and experimentally that models built from exponentially large quantum feature space can be classically reproduced by sampling a few frequencies to build an equivalent low dimensional kernel. Precisely, we show that the number of required samples grows favorably with the size of the quantum spectrum. This tool therefore questions the hope for quantum advantage from VQCs in many cases, but conversely helps to narrow the conditions for their potential success. We expect VQCs with various and complex encoding Hamiltonians, or with large input dimension, to become more robust to classical approximations.

## Introduction

Many applications of quantum computing in the near term rely on variational quantum circuits (VQCs) (Bharti et al. (2021); Cerezo et al. (2021)), and in particular for solving machine learning (ML) tasks. VQCs are trained using classical optimization of their gates' parameters, a method borrowed from classical neural networks. Although the term VQCs encompasses various methods for different ML tasks, in this paper we refer to VQCs as a family of quantum models for supervised learning on classical data, as defined in B.1. Many early works have shown promising results, both empirically and in theory (Cong et al. (2019); Huang et al. (2021)). However, whether these variational methods can provide a quantum advantage in the general case with a scaling number of qubits has yet to be proved.

The notion of quantum advantage with respect to classical methods is plural and can be defined as advantage in trainability (Larocca et al. (2021)), expressivity (Schuld et al. (2021)) or generalization (Caro et al. (2022)). In this paper, we focus on the expressive power of VQCs as it is crucial to understand the capacity of VQCs to generate models that would be hard to do with classical approaches. The most common and intuitive answer for expressive power of VQCs is the formation of a *large feature space*, due to the projection of data points in an exponentially large Hilbert space. Understanding what is happening in this Hilbert space, and most importantly, knowing how we can exploit its size is an important question for this field. Regarding expressivity, Fair comparison

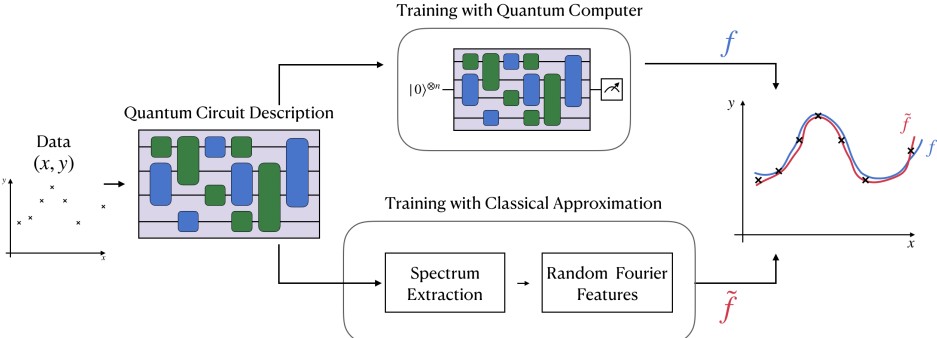

Figure 1: **Random Fourier Features as a classical approximator of quantum models.** Instead of training a Variational Quantum Circuit by using a quantum computer, we propose to train a classical kernel built by sampling a few frequencies of the quantum model. These frequencies can be derived from the quantum circuit architecture, in particular from the encoding gates. Using Random Fourier Features, one can build a classical model which performs as good as the quantum model with a bounded error and a number of samples that grows nicely.

between VQCs and classical ML algorithms is a crucial challenge. Recent works (Schuld et al. (2021); Schuld (2021); Schreiber et al. (2022)) have shown equivalence between VQCs and both Fourier series and kernel methods. In fact, VQCs are powerful Fourier series: the functions they can learn are predetermined by a set of frequencies which can become numerous with the dimension of the input, and the number and complexity of the quantum gates used.

In this work, we adapt Random Fourier Features (RFF) from Rahimi & Recht (2009), a classical sampling algorithm aimed at efficiently approximating some large classical kernel methods. We design three different RFF strategies to approximate VQCs. For each one, we analyze its efficiency in reproducing results obtained by a VQC. To do so, we study in detail the expressivity power of VQCs and compare it each time with RFF.

Our method consists in analyzing the encoding gates of the VQC to extract the final frequencies of its model and sample from them. Notably, if the VQCs possesses simple encoding gates such as Pauli gates (e.g. $R_Z$), we show that the large quantum feature space is not fully exploited, making RFF even more efficient. If the number of frequencies in the VQC grows exponentially, the number of samples required by RFF grows only linearly. Finally, we have empirically compared VQCs and RFF on real and artificial use cases. On these, RFF were able to match the VQC's answer, and sometimes outperform it.

As a whole, the novelty of our work is to identify cases where the quantum model emanating from a VQC can be classically approximated, albeit the VQC being non-classically simulatable. We therefore reduce the hope for expressive advantage of VQCs in such cases. In practice, our method is limited by several parameters: inputs with very large dimension, encoding Hamiltonians that are hard to diagonalize, large depths and others (see Appendix F). We can therefore use the limitations of our method as a recipe to build VQCs that are resilient to classical approximation.

## 1 SPECTRAL PROPERTIES OF VARIATIONAL QUANTUM CIRCUITS (VQCS)

In this Section, we succinctly recall some properties of VQCs in the context of machine learning, which will be useful for this work. We invite readers that are not familiar with VQCs to consult instead the detailed definitions and explanations provided in the Appendix B.

VQCs are quantum circuits with parametrized gates. In this paper the *encoding* gates are unitary matrices of the form $\exp(-ix_j H)$, where $H$ is a Hamiltonian that the quantum computer can implement, and $x = (x_1, \cdots, x_d) \in \mathbb{R}^d$ is some input data. The other class of gates are *trainable* gates and use parameters $\theta$ that are optimized during the learning phase. At the end of the circuit, measurements are performed and yield a classical output called the quantum model $f$.

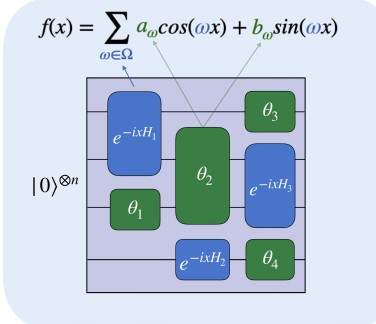
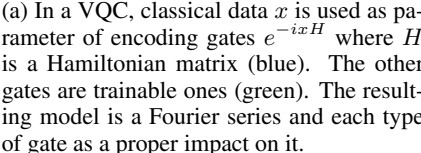
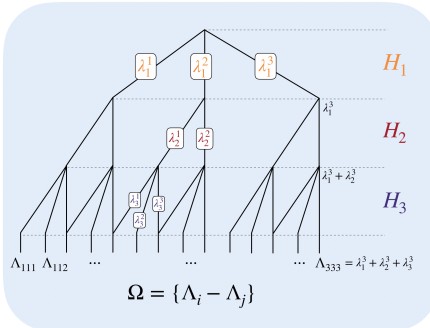

(a) In a VQC, classical data $x$ is used as parameter of encoding gates $e^{-ixH}$ where $H$ is a Hamiltonian matrix (blue). The other gates are trainable ones (green). The resulting model is a Fourier series and each type of gate as a proper impact on it.

(b) The frequencies composing the VQC model come from all the combinations of eigenvalues $\lambda_i^j$ from each encoding Hamiltonian $H_j$, seen as a tree. Redundancy or concentration in the leaves is common if the eigenvalues are similar.

Figure 2: Variational Quantum Circuits (1-dimensional example) give rise to Fourier Series *(a)* and its frequencies can be derived from the Hamiltonians of encoding gates *(b)*.

With one-dimensional input ($x \in \mathbb{R}$), as shown in Figure 2, this function can be expressed as a discrete Fourier series $f(x; \theta) = \sum_{\omega \in \Omega} a_\omega \cos(\omega x) + b_\omega \sin(\omega x)$ (Schuld et al. (2021)). We understand that the spectrum $\Omega$ will be the key element that defines the potential expressivity of $f$. The frequencies $\omega \in \Omega$ are in fact derived from the eigenvalues $\{\lambda_i^j\}_i$ of each encoding Hamiltonian $H_j$. With $L$ encoding gates, we have $\Omega = \left\{ \Lambda_{\boldsymbol{i}} - \Lambda_{\boldsymbol{j}}, \boldsymbol{i}, \boldsymbol{j} \in [N]^L \right\}$, where $\Lambda_{\boldsymbol{i}} = \lambda_1^{i_1} + \cdots + \lambda_L^{i_L}$ are a sum of eigenvalues, one per Hamiltonian (see Figure 2).

With $d$-dimensional input, $\Omega$ is constructed as the Cartesian product over the spectrum of each dimension, therefore $\Omega$ is a set of $d$-dimensional frequencies.

We will sometimes focus on Pauli encoding gates, where all encoding Hamiltonians are simple Pauli matrices, with two eigenvalues $\lambda = \pm 1$. In this case, and with 1-dimensional inputs, we have $\Omega = [\![0, L]\!]$, the set of positive integer frequencies. In dimension $d$, the spectrum's size becomes $|\Omega| = O(L^d)$ (see Eq.11 for details). A scaled version of Pauli encoding exists, where the Pauli are multiplied with a factor as in Kyriienko et al. (2022); Shin et al. (2022), resulting in larger $\Omega$.

We can also write the quantum model in the form $f(x; \theta) = \mathbf{w}(\theta)^T \phi(x)$ (Schuld (2021)), where

$$\phi(x) = \frac{1}{\sqrt{|\Omega|}} \begin{bmatrix} \cos(\omega^T x) \\ \sin(\omega^T x) \\ \vdots \end{bmatrix}_{\omega \in \Omega} \quad \text{is the } \textit{feature } \text{vector, and} \quad \mathbf{w}(\theta) = \begin{bmatrix} a_\omega \\ b_\omega \\ \vdots \end{bmatrix}_{\omega \in \Omega} \quad \text{is the } \textit{weight } \text{vector.}$$

$\phi(x)$ is a mapped version of the input in a space of dimension $2|\Omega|$. An exponential size for $\Omega$ could be a key argument to the quantum advantage of VQC. We define the associated kernel as $k(x, x') = \phi(x)^T \phi(x')$. In Appendix B.1, we show that $k$ has a *shift-invariance* property, which will be useful later on, when using the Random Fourier Feature framework.

## 2  RANDOM FOURIER FEATURES APPROXIMATES HIGH-DIMENSIONAL KERNELS

In this Section, we explain the key results of the classical method called Random Fourier Features (RFF) (Rahimi & Recht (2009); Li et al. (2019); Sutherland & Schneider (2015)). We will use this method to create several classical sampling algorithms for approximating VQCs.

Let $\mathcal{X} \subset \mathbb{R}^d$ be a compact domain and $k : \mathcal{X} \times \mathcal{X} \longrightarrow \mathbb{R}$ be a kernel function. We assume $k$ is shift invariant, meaning $k(x, y) = \overline{k}(x - y) = \overline{k}(\delta)$, where $\overline{k} : \mathcal{X} \longrightarrow \mathbb{R}$ is a single variable function, and

we will note $\overline{k} = k$ to simplify the notation. From Rudin (2017), Bochner's theorem insures that the Fourier transform of $k$ is a positive function and we can write $k(\delta) = \int_{\omega \in \mathcal{X}} p(\omega) e^{-i\omega^T \delta} d\omega$. If we assume $k$ is also normalized, then the Fourier transform $p(\omega)$ of $k$ can be assimilated to a probability distribution. With a dataset of $M$ points, fitting a Kernel Ridge Regression (KRR) model with the kernel $k$ necessitates $M^2$ operations to compute the kernel matrix and $\mathcal{O}(M^3)$ to invert it. This becomes impractical when $M$ reaches high value in modern big datasets.

The idea of the Random Fourier Feature method (Rahimi & Recht (2009)) is to approximate the kernel $k$ by $\tilde{k}(x, y) \simeq \tilde{\phi}(y)^T \tilde{\phi}(x)$, where $\tilde{\phi}(x) = \frac{1}{\sqrt{D}} \begin{bmatrix} cos(\omega_i^T x) \\ sin(\omega_i^T x) \\ \vdots \end{bmatrix}_{i \in [\![1,D]\!]}$ , where the $\omega_i$s are $D$ frequencies *i.i.d* sampled from the frequency distribution $p(\omega)$. Formally, it is a Monte-Carlo estimate of $k$. Note that $p(\omega)$ can be analytically found in some cases such as Gaussian or Cauchy kernels (Rahimi & Recht (2009)). Then instead of fitting a KRR for $k$, one will solve a Linear Ridge Regression (LRR) with $\phi$ (see details in C). The two problems are equivalent (Bishop & Nasrabadi (2006)), and the number of operations needed for the LRR is $\mathcal{O}(MD^2 + D^3)$. If $D$ is much smaller than $M$, it is much faster than solving the KRR directly. Even if $D$ is so big that the linear regression cannot be exactly solved, one can employ stochastic gradient descent or adaptive momentum optimizers such as Adam (Kingma & Ba (2014)). The output of the LRR or gradient descent is simply a weight vector $\tilde{\mathbf{w}}$ that is used to create the approximate function $\tilde{f} = \tilde{\mathbf{w}}^T \tilde{\phi}(x)$.

In Sutherland & Schneider (2015), authors bound the resulting approximation error in the RFF method. We provide the main theorems in Appendix D. In Theorem 3, we consider the error between the final functions $f$ and $\tilde{f}$. We see that if the error must be constrained such that $|f(x) - \tilde{f}(x)| \leq \epsilon$, for $\epsilon > 0$, one can derive a lower bound on the number $D$ of necessary frequency samples for the approximation to hold. Fortunately, the bound on $D$ grows linearly with the input dimension $d$, and logarithmically with $\sigma_p$ which is linked to the variance of the frequency distribution $p(\omega)$.

Finally, note that it is important to take into account the Shannon criterion, stating that one needs at least $2\omega_{max}$ training points to estimate the coefficients of a Fourier series of maximum frequency $\omega_{max}$. In practice, it puts some limitation on the largest frequency one can expect to learn (both classically and quantumly) given an input dataset. Exponentially large frequencies with VQCs, as in Kyriienko et al. (2022); Shin et al. (2022) would have a limited interest with subexponential number of training points. The efficiency of RFF against VQCs in such cases becomes even more interesting, as it allows reducing the number $D$ of sample compared to the actual exponential size of $\Omega$.

## 3 RFF METHODS FOR APPROXIMATING VQCS

In this Section, we present our solutions to approximate a VQC model using classical methods. The intuitive idea is to sample some frequencies from the VQC's frequency domain $\Omega$, and train a classical model from them, using the RFF methods from Section 2. We present different strategies to sample these frequencies and detail in which cases we expect our methods to perform well. In Appendix F, we study potential limitations to our methods, opening the way for VQCs with strong advantage over classical methods.

### 3.1 RFF WITH DISTINCT SAMPLING

This strategy described in Algorithm 1 is the basic approach of using RFF for approximating VQCs. We assume the ability to sample from $\Omega$. It is straightforwardly following the method described in Section 2. Indeed, as explained in Section 1, the corresponding kernel $k$ of the VQC is built from frequencies $\omega \in \Omega$ and is shift-invariant (see Appendix B.3), which ensures us the efficiency of RFF to approximate them. Applying the RFF method would consist in sampling from $\Omega$ and reconstructing a classical model to approximate the VQC output function $f$. As shown in Theorem 1, one might require a number $D$ of samples for which a lower bound scales linearly with the dimension $d$ of the input, and logarithmically with the size of $\Omega$.

---

**Algorithm 1** RFF with Distinct Sampling

---

**Require:** a VQC model $f$, and $M$ data points $\{x_j\}_{j \in [M]}$

**Ensure:** Approximate function $\tilde{f}$

1: Diagonalize the Hamiltonians of the VQC's encoding gates.
2: Use their eigenvalues to obtain all frequencies $\omega \in \Omega$, as in Eq.7
3: Sample $D$ frequencies $(\omega_1, \cdots, \omega_D)$ from $\Omega$
4: Construct the approximated kernel $\tilde{k}(x, y) = \tilde{\phi}(y)^T \tilde{\phi}(x)$ with $\tilde{\phi}(x) = \frac{1}{\sqrt{D}} \begin{bmatrix} cos(\omega_i^T x) \\ sin(\omega_i^T x) \end{bmatrix}_{i \in [\![1, D]\!]}$
5: Solve the LRR problem (Appendix C), and obtain a weight vector $\tilde{\mathbf{w}}$.
6: Obtain the approximated function $\tilde{f}(x) = \tilde{\mathbf{w}}^T \tilde{\phi}(x)$

---

## 3.2 RFF with Tree sampling

The abovementioned method requires constructing explicitly $\Omega$, which can become exponentially large if the dimension $d$ of the data points is high (see Appendix B.2). The size of $\Omega$ is also increased if the encoding Hamiltonians have many eigenvalues (like many-body Hamiltonians) or if many encoding gates are used. A large $\Omega$ is indeed the main interest of VQCs in the first place, promising a large expressivity.

In some cases, and in particular for a VQC using many Pauli encoding gates, a lot of redundancy occurs in the final frequencies. Indeed, if many eigenvalues are equal, the tree leaves will become redundant. Very small eigenvalues can also create groups of frequencies extremely close to each other, which in some use cases, when tuning their coefficients $a_\omega$ and $b_\omega$, can be considered as redundancy. In our numerical experiments (see Figure 7), we observe an interesting phenomenon: on average, the frequencies with the more redundancy tend to obtain larger coefficients. Conversely, isolated frequency are very likely to have small coefficients in comparison, making them "ghost" frequencies of $\Omega$. For VQC with solely Pauli encoding gates, we observe that the coefficients of high frequencies are almost stuck to zero during training. Therefore, one can argue that the *Distinct Sampling* described above can reach even more frequencies than the corresponding VQC. However, if one wants to closely approximate a given VQC with RFF, one would not want to sample such isolated frequencies from $\Omega$ but instead draw with more probability the redundant frequencies.

---

**Algorithm 2** RFF with Tree Sampling

---

**Require:** a VQC model $f$, and $M$ data points $\{x_j\}_{j \in [M]}$

**Ensure:** Approximate function $\tilde{f}$

1: Diagonalize the Hamiltonians of the VQC's encoding gates.
2: Sample $D$ paths from the tree shown in Figure 2, obtain $D$ frequencies $(\omega_1, \cdots, \omega_D)$ from $\Omega$
3: Follow steps 4-6 of Algorithm 1.

---

This is what we try to achieve with *Tree Sampling*. Knowing the eigenvalue decomposition of each encoding's Hamiltonian, we propose to directly sample from the tree shown in Figure 2. The first advantage of this method is that it does not require computing the whole set $\Omega$, but only draw $D$ paths through the tree (which can be used to generate up to $\binom{D}{2} + 1$ positive frequencies, with potential redundancy). Secondly, it naturally tends to sample more frequencies that are redundant, and therefore more key to approximate the VQC's function. Overall, it could speed up the running time and require fewer samples.

## 3.3 RFF with Grid sampling

The two above methods suffer from a common caveat: if one or more of the encoding Hamiltonians are hard to diagonalize, sampling the VQC's frequencies is not possible as it prevents us from building some of the branches of the tree shown in Figure 2.

Even in this case, we propose a method to approximate the VQC. If the frequencies are unknown, but one can guess an upper bound or their maximum value, we propose the following strategy: We create a grid of frequencies regularly disposed between zero and the upper bound $\omega_{\max}$, on each

dimension. In practice, if unknown the value of $\omega_{\max}$ can simply be the largest frequency learnable by the Shannon criterion (see Section 3.4) hence half the number of training points. Letting $s > 0$ be the step on this grid, the number of frequencies on a single dimension is given by $\omega_{\max}/s$. Over all dimensions, there are $\lceil(\omega_{\max}/s)\rceil^d$ frequency vectors. Therefore, instead of sampling from actual frequencies in $\Omega$, one could sample blindly from this grid, hence the name *Grid Sampling*. At first sight, it might seem ineffective, since none of the frequencies actually in $\Omega$ may be represented in the grid. But we show in the Appendix E.2 (Theorem 4) that the error between the VQC's model $f$ and the approximation $\tilde{f}$ coming from the grid can be bounded by $s$. When $s$ is small enough, the number $D$ of samples to reach an error $\epsilon > 0$ grows like $1/\epsilon^2 log(1/s)$ which is surprisingly efficient. However, the trade-off comes from the fact that a small $s$ means a very large grid, in particular in high dimension.

---

**Algorithm 3** RFF with Grid Sampling

---

**Require:** Assumption on the highest frequency $\omega_{\max}$, a step $s > 0$ and $M$ data points $\{x_j\}_{j\in[M]}$

**Ensure:** Approximate function $\tilde{f}$
 1: Create a regular grid in $[0, \omega_{\max}]^d$ with step $s$.
 2: Sample $D$ frequencies $(\omega_1, \cdots, \omega_D)$ from the grid.
 3: Follow steps 4-6 of Algorithm 1.

---

### 3.4 NUMBER OF SAMPLES AND APPROXIMATION ERROR

As introduced in Section 2 (and detailed in Theorem 3), the quality of the approximation of RFF to a certain error $\epsilon > 0$ depends on $p(\omega)$, its variance, and the number $D$ of frequency samples taken from it. In our case, the continuous distribution $p(\omega)$ will be replaced by the actual set of frequencies $\Omega$, such that $p(\omega) = \frac{1}{|\Omega|} \sum_{\omega\in\Omega} \delta_\omega$ , where $\delta_\omega$ represents the Dirac distribution at $\omega$. As a result, we can write the discretized variance as $\sigma_p = \sum_{\omega\in\Omega} p(\omega)\omega^T\omega$. From this, we want to know the link between the number $D$ of necessary samples and the size of $\Omega$, or even the number $L$ of encoding gates per dimension. In the general case, we consider that $\sigma_p$ is the average value of $\omega^T\omega$, that is to say the trace of the multidimensional variance. The more the frequencies are spread, the higher $\sigma_p$ will be, but the number of frequencies in itself doesn't seem to play the key role here.

We provide here a bound on the minimum of samples required to achieve a certain error between the RFF model and the complete model in the case of Pauli encoding in the distinct sampling strategy. The proof and details for this theorem is shown in Appendix Section E.1.

**Theorem 1.** *Let $\mathcal{X}$ be a compact set of $\mathbb{R}^d$, and $\epsilon > 0$. We consider a training set $\{(x_i, y_i)\}_{i=1}^M$. Let $f$ be a VQC model with $L$ encoding Pauli gates on each of the $d$ dimensions, and full freedom on the associated frequency coefficients, trained with a regularization $\lambda$. Let $\sigma_y^2 = \frac{1}{M} \sum_{i=1}^M y_i^2$ and $|\mathcal{X}|$ the diameter of $\mathcal{X}$. Let $\tilde{f}$ be the RFF model with $D$ samples in the distinct sampling strategy trained on the same dataset and the same regularization. Then we can guarantee $|f(x) - \tilde{f}(x)| \leq \epsilon$ with probability $1 - \delta$ for a number $D$ of samples given by:*

$$D = \Omega\left(\frac{dC_1(\lambda + 1)^2}{\lambda^4\epsilon^2}\left[log(dL^2|\mathcal{X}|) + log\frac{C_2(\lambda + 1)}{\epsilon\lambda^2} - log\delta\right]\right) \tag{1}$$

*with $C_1$ and $C_2$ being constants depending on $\sigma_y$, $|\mathcal{X}|$. We recall that in Eq.1 the notation $\Omega$ stands for the computational complexity "Big-$\Omega$" notation.*

We can conclude that the number $D$ of samples grows linearly with the dimension $d$, and logarithmically with the size of $\Omega$. It means that even though the number of frequencies in the spectrum of the quantum circuit is high, only a few of them are useful in the model. This fact limits the quantum advantage of such circuits. However, the scaling in $\epsilon$ and $\lambda$, respectively in $\Omega(1/\epsilon^2)$ and $\Omega(1/\lambda^4)$ is not favorable, and can limit in practice the use of the RFF method.

An important clarification has to be made over the interpretation of this result. One may think at first sight that the RFF method would be efficient to approximate the outputs of any VQC, or find

a "classical surrogate" (Schreiber et al. (2022)). Instead, the bound that is provided is on the error between a VQC trained on an independent data source and a RFF model trained on the same data. If the dataset fails to correctly represent the data distribution, then the VQC will fail to correctly model it, and the theorem provides the minimal number of samples to perform "as badly", and this number could be low. This was tested in the numerical simulations in Section 4. However, we recall that these are bounds that we cannot reach in practice with the current classical and quantum computing resources. They give an intuition on the asymptotic scaling as quantum circuits become larger. Note that this proof changes if we take into account the redundancy of each frequency when sampling from $\Omega$. This will be the case in the *Tree Sampling* strategy. In that case, the variance becomes even smaller since some frequencies are more weighted than others, in particular for Pauli encoding.

In Appendix E.2 we provide a similar theorem for the *Grid* sampling strategy and general encoding Hamiltonians, summarized in Theorem 4.

## 4 EXPERIMENTS AND NUMERICS

In this Section, we aim to assess the accuracy and efficiency of our classical methods to approximate VQCs in practice. Each VQC was analyzed using ideal simulators of quantum computers, on a classical computer, without taking the noise into account. Important complementary experiments are provided in Appendix G. In particular, we show scaling simulations in Appendix G.5.

### 4.1 USING RFF TO MIMIC RANDOM VQC

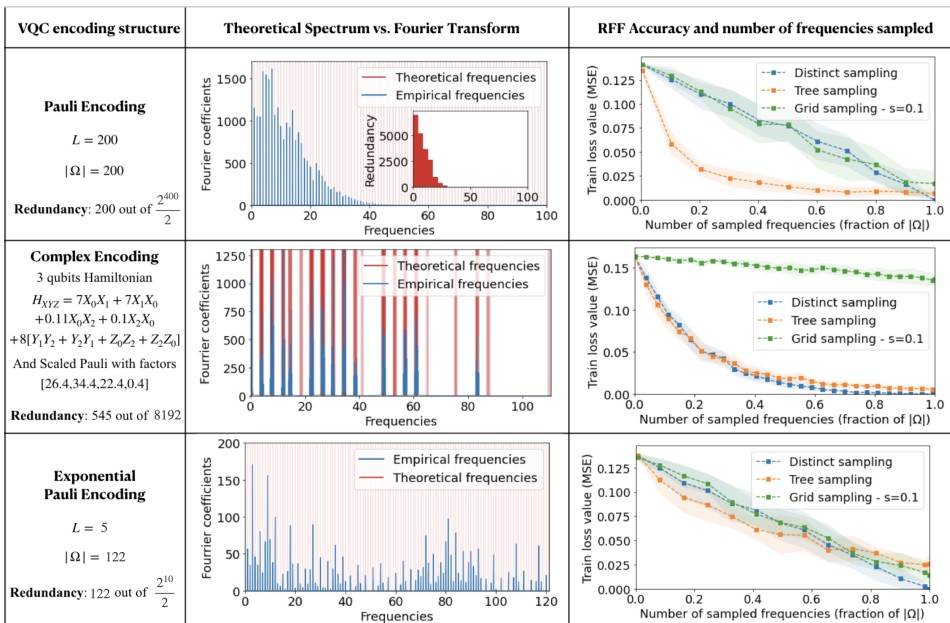

Figure 3: **Approximating random VQCs with three RFF methods.** For each type of encoding (basic Pauli, many-body Hamiltonian, scaled Pauli) we have: *(Left)* averaged the VQC's model and compared its empirical Fourier Transform (blue lines) to the expected Spectrum $\Omega$ (red lines), and *(Right)* measured how many frequency samples were necessary to reproduce the same model classically with our three RFF methods.

In a first stage, we focus on approximating random VQCs (i.e. with random parameter initialization) using Random Fourier features. All results are gathered in Figure 3. To this end, we fix the quantum spectrum $\Omega$ by making a certain choice about the structure of the encoding gates. We then observe and evaluate the performance of our three RFF strategies (*Distinct*, *Tree*, and *Grid* sampling to approximate the quantum model. For completeness, we have tested our methods on several types of VQCs : some with basic Pauli encoding in 1 dimension, in higher dimension (Figure 8) , with non-trivial encoding Hamiltonians, and with scaled Pauli encoding as in Shin et al. (2022). For each

type, we have also observed the actual Fourier Transform of the random VQCs model on average, to understand which frequencies appear more frequently in their spectrum. We recommend reading Appendix G.2 for details on our methodology and interpretation.

With these experiments, we conclude a few important properties. We observe that when some frequencies in the spectrum $\Omega$ have much redundancy (*e.g.* Pauli encoding), these frequencies are empirically the ones with higher coefficients. In such case, the *Tree* sampling strategy is able to approximate the VQC's model with fewer samples than the other methods as expected. With more complex Hamiltonians (see Appendix G.2.2 for details), concentrated packets of frequencies appear, and even without much redundancy, both *Tree* and *Distinct* sampling require fewer frequency samples to cover these packets. According to these experiments, the worst case scenario for the RFF is a uniform probability distribution where all the three sampling techniques will be equivalent. Nonetheless, the theoretical bounds prove that the number of Fourier Features will scale linearly with respect to the spectrum size that scales itself exponentially.

## 4.2 Comparing VQC and RFF on Artificial Target Functions

A more practical comparison is to measure the efficiency of both VQC and RFF on a common target function. We want to see when RFF can obtain a similar or better results than the VQC on the same task, trained separately. We have seen that for VQCs with Pauli encoding, the Fourier coefficients are rapidly decreasing, cutting out frequencies higher than $\omega_{\text{effective}}$ from the empirical spectrum. For this reason, we have chosen a particular synthetic target function: we create a sparse Fourier series as a target function $s(x)$ and a VQC with $L = 200$ Pauli encoding gates as the quantum model. In Figure 4, we clearly see that the VQC as well as RFF with *Tree* sampling cannot learn the frequency $\omega = 60 > \omega_{\text{effective}}$ (their train loss reach a high limit) while the RFF models based on *Distinct* and *Grid* sampling can effectively learn the target function with enough frequency samples. This result shows that even when a VQC with Pauli encoding is trained, it cannot reach all of its theoretical spectrum, thus questioning the expressivity of such a quantum model, whereas its classical RFF approximator succeeds in learning the function in question. Of course, when we chose an artificial function for which all frequencies are below $\omega_{\text{effective}}$, the VQC and all RFF manage to fit it.

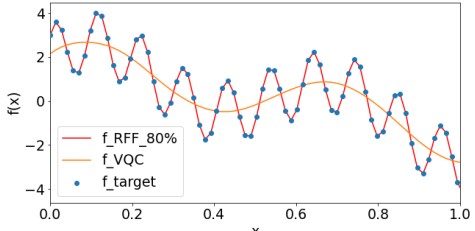
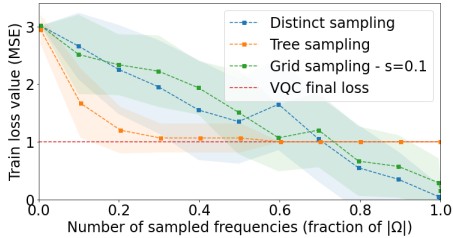

(a) Predictions of the target function $s(x)$ with the quantum model and its corresponding RFF approximator using *Distinct* sampling on 80% of all frequencies.

(b) RFF learning curves for sparse target fitting based on the VQC description. Distinct and Grid sampling are able to outperform the VQC.

Figure 4: **Fitting a target function** $s(x) = \sum_{\omega \in \{4,10,60\}} cos(\omega x) + sin(\omega x)$ with a VQC architecture of $L = 200$ Pauli gates.

## 4.3 Comparing VQC and RFF on Real Datasets

Here again, we train separately a VQC and the equivalent RFF model on real-world data to compare their learning performances We choose the fashion-MNIST dataset (Xiao et al. (2017)), where we consider a binary image classification task. We also use the California Housing dataset for a regression task. We chose to solve the two problems by training VQCs with Pauli encoding ($L = 5$ for each dimension). According to Eq.11, the number of distinct positive frequency in $\Omega$ is 80526. In Figure 5, we observe that, with very few frequencies sampled, the RFF model with *Tree* sampling succeeded in learning the distribution of the input datasets as well as closely approximating the trained quantum model with a radically lower number of frequencies ($0.002 \times |\Omega| \simeq 160$). This result could indicate that the underlying distributions to learn were simple enough, such that the VQC

had an excessive expressivity but could also point to the regularization power of a VQC potentially emerging from redundancies (Peters & Schuld, 2022).

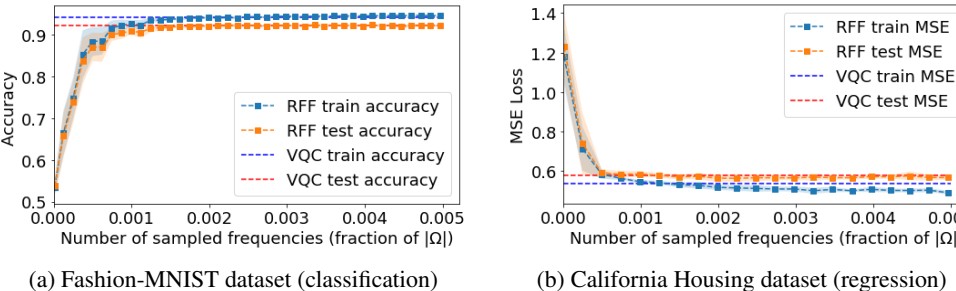

(a) Fashion-MNIST dataset (classification)  (b) California Housing dataset (regression)

Figure 5: Prediction results of RFF (Tree Sampling) on 5-dimensional real datasets.

## 4.4 NUMERICAL EXPERIMENTS DISCUSSION

The simulations confirm that the spectrum of VQCs are predictable from their encoding gates. We observe however that the actual set of frequencies that emerge with non-zero coefficients does not cover the whole spectrum, questioning the effective expressivity of VQCs in practice. It also confirms the intuition that frequencies that appear with high redundancy in the spectrum tend to have larger coefficients. The three RFF alternatives we propose to approximate VQCs on several tasks revealed to be efficient. The number $D$ of samples (seen as a fraction of the spectrum size) grows favorably and allows for good approximation. With a highly redundant spectrum our *Tree Sampling* method, which is the most computationally efficient, was able to approximate VQCs with fewer frequency samples. Tree Sampling inherits the same drawbacks as VQCs and are sometimes not capable of learning less redundant frequencies in the VQCs spectrum, whereas *Distinct* and *Grid* sampling can outperform the VQCs in such cases. Even if these experiments are encouraging and match our expectations, we are far from the maximal efficiency of RFF methods. Indeed, the scaling of $D$, the number of samples, becomes even more favorable when the input dimension $d$ increases, or more generally when the spectrum has an exponential size. We expect to get closer to the theoretical bounds and see the number of samples becoming an even smaller fraction of the size of the spectrum. This was partially confirmed in our small scale simulations up to $d < 7$. In that regime the theoretical bounds blow up because of the factor in $1/\lambda^4$ so they are not relevant. We have seen that, in order to match the VQCs, some RFF required using 80% of the frequencies, other only 0.001%. We could go further by training VQCs on actual quantum computers and compare them to classical RFF methods.

## 5 CONCLUSION

In this work, we have studied the potential expressivity advantage of Variational Quantum Circuits (VQCs) for machine learning tasks, by providing novel classical methods to approximate VQCs. Our three methods use the fact that sampling few random frequencies from a large dimensional kernel (exponentially large in the case of VQCs) can be enough to provide a good approximation. This can be done given only the description of the VQCs and does not require running it on a quantum computer. We studied in depth the number of samples and its dependence on key aspects of the VQCs (input dimension, encoding Hamiltonians, circuit depth, VQC spectrum, number of training points). On the theoretical side, we conclude that our classical sampling method can approximate VQCs for machine learning tasks on Hamiltonian encoding of classical data, with a number of samples that scales favorably but with potentially large constant overheads. Experimentally, we have tested our classical approximators on several use cases, using both artificial and real datasets, and our classical methods were able to match or exceed VQC results. By providing a new way of comparing classical and quantum models, these results may help to understand where the true power of VQCs comes from, and define more rigorously quantum advantage in this context. In particular, it opens up questions about alternative encoding schemes, harnessing the effective expressivity of VQCs, and the link between a full expressivity and trainability.

## 6 ACKNOWLEDGEMENT

This work was supported by the Engineering and Physical Sciences Research Council (grants EP/T001062/1) and the H2020-FETOPEN Grant PHOQUSING (GA no.: 899544).

## 7 REPRODUCIBILITY STATEMENT

We hereby state that the results of this paper are reproducible. Theory-wise, the shift-invariance of the quantum kernel is proved in Appendix B.3, the approximation results for Random Fourier Features are presented and proven in Appendix D and the approximations results for Random Fourier Features in the context of Variational Quantum Circuits are presented in the Appendix E. For numerical simulations, methods to generate random VQCs are detailed in Appendix G.2. The code used is available in the supplementary materials. The anonymous link is also provided here `https://osf.io/by5dk/?view_only=5688cba7b13d44479f76e13e01d28d75`

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

## A RELATED WORK

A recent work Schreiber et al. (2022) independently proposed a similar approach where classical surrogate methods approximate VQCs. The difference with this work is the necessity of having access to all $\Omega$, the totality of the frequencies of the VQC considered, without sampling from them.

Indeed, if $\Omega$ is known, the coefficients $a_\omega$ and $b_\omega$ of the VQC function (see Eq.10) can be easily fitted by solving the classical least square problem. Namely, one determines $\mathbf{w}^*$ such that

$$\mathbf{w}^* = \arg\min_{\mathbf{w}} \frac{1}{M} \sum_{i=1}^{M} |\mathbf{w}^T \phi(x_i) - y_i|^2 + \lambda_0 ||\mathbf{w}||^2 \tag{2}$$

where $\phi(x) = \begin{bmatrix} cos(\omega x) \\ sin(\omega x) \end{bmatrix}_{\omega \in \Omega}$, and $\lambda_0$ is the regularisation parameter. As explained in the previous Section, with a dataset of $M$ points, this can be solved exactly using matrix inversion in $\mathcal{O}(M|\Omega|^2 + |\Omega|^3)$ operations if $M \geq 2|\Omega|$. If the inequality is not fulfilled or if $|\Omega|^3$ is too big, one would use stochastic gradient descent instead of matrix inversion.

However, this method assumes that $\Omega$, and is not too large, which will usually be the case as we shown in section B.2. One should also be able to enumerate all individual frequencies $\omega \in \Omega$. Moreover, as we will show the redundancy of some frequencies in $\Omega$ has a key importance, which is not captured by such a method.

For completeness, we note from the seminal work Schuld (2021) that the author briefly mentions the idea of approximating kernels with RFF. Similarly, in the recent work Peters & Schuld (2022), the authors mention RFF as a sampling strategy on VQCs with shift invariant kernels, without further details. In Gyurik & Dunjko (2022), the authors mention briefly that VQCs with polynomially large Fourier series can be learned from a polynomial number of training points, but this method remain unclear for exponentially large Fourier series.

We also note that other works (Yamasaki et al. (2020); Nakaji et al. (2022)) make use of random features, for or from quantum circuits, but are less related to our approach.

## B PRELIMINARIES ON VARIATIONAL QUANTUM CIRCUIT FOR MACHINE LEARNING

### B.1 DEFINITIONS

We provide here a definition of VQCs that is used in this work. Many more models in the quantum machine learning litterature such as quantum circuit born machines (Liu & Wang, 2018), variational imaginary time evolution (Zoufal et al., 2021), differentiable quantum circuits (Kyriienko et al., 2021), quantum evolution kernel (Henry et al., 2021), quantum orthogonal neural networks (Kerenidis et al., 2021), to cite only a few, are also variational but we don't consider them here. It is an open question to extend our findings to such models.

We consider a standard ML task where a function $f$, named *model*, must be optimized to map data points to their target values. The data used of the training is made of $M$ points $x = (x_1, \ldots, x_d)$ in $\mathcal{X} = \mathbb{R}^d$ along with their target values $y$ in $\mathcal{Y} = \mathbb{R}$. We define a *quantum model* as the family of parametrized functions $f : (\mathcal{X}, \Theta) \longrightarrow \mathcal{Y}$, such that

$$f(x; \theta) = \langle 0|U(x; \theta)^\dagger O U(x; \theta)|0\rangle \tag{3}$$

where $U(x; \theta)$ is a unitary that represents the parametrized quantum circuits, $\theta$ represents the trainable parameters from a space $\Theta$, and $O$ is an observable. The parameterized quantum circuit is

composed of layers of alternating *embedding* and *trainable* unitaries. Embedding unitaries depend on input data values whereas trainable unitaries depend on internal parameters that are optimized during training. A typical instance of a VQC is illustrated in Figure 2. Note that an actual layer structure is not mandatory, since any circuit can be sliced into alternating sequences of encoding and training blocks (even if containing a single gate).

Any quantum unitary implements the evolution of a quantum system under a Hamiltonian. Thus, we choose to write the $\ell^{th}$ encoding gates as $\exp(-ix_iH_\ell)$, where $x_i$ is one of the $d$ components of $x$, and $H_\ell$ is a Hamiltonian matrix of size $2^p$ if $p$ is the number of qubits this gates acts on. We will note $L$ the number of encoding gates for each dimension of $x$ (the same for each dimension, for notation simplicity).

In this framework, the aim is to find the optimal mapping between data points and their target values. This is done by optimizing the parameters $\theta$ to find the best guess $f^*$ such that

$$f^* = \arg\min_\theta \frac{1}{M} \sum_{i=1}^{M} l(f(x_i; \theta), y_i) \tag{4}$$

where $l$ is a cost function adapted to the task. For a standard regression tasks, we can choose $l(z, y) = |z - y|^2$.

## B.2 QUANTUM MODELS ARE LARGE FOURIER SERIES

We know since Schuld et al. (2021) that the family of quantum models defined in Eq.(3) can be rewritten as a Fourier series:

$$f(x; \theta) = \sum_{\omega \in \Omega} c_\omega e^{i\omega x} \tag{5}$$

where the spectrum $\Omega$ of frequencies is determined by the ensemble of eigenvalues of the encoding Hamiltonians and the coefficients $c_\omega$ depend on the parametrized ansatz, as pictured in Figure 2.

In order to familiarize the reader with the structure of the spectrum, we explicitly build $\Omega$ in the case of a one dimensional data input ($\mathcal{X} = \mathbb{R}$) and with a variational circuit containing only $L$ encoding gates. The accessible frequency spectrum $\Omega$ is the ensemble of all the differences between all possible sums of the eigenvalues of the encoding gates as shown in Figure 2.

We note $\lambda_\ell^k$ the $k^{th}$ eigenvalue of the $\ell^{th}$ encoding Hamiltonian $H_\ell$. We use the multi-index $\boldsymbol{i} = (i_1, \ldots, i_L)$ indicating which eigenvalue is taken from each encoding Hamiltonian. We define $\Lambda_{\boldsymbol{i}}$ as

$$\Lambda_{\boldsymbol{i}} = \lambda_1^{i_1} + \cdots + \lambda_L^{i_L} \tag{6}$$

Finally, we can express the set of frequencies as:

$$\Omega = \left\{ \Lambda_{\boldsymbol{i}} - \Lambda_{\boldsymbol{j}}, \boldsymbol{i}, \boldsymbol{j} \in [N]^L \right\}, \tag{7}$$

*Proof.* We consider a circuit unitary of the following form $U(x; \theta) = W^{(L+1)}S^{(L)}(x)\ldots S^{(1)}(x)W^{(1)}$ as described in B.1 where $W^{(l)}$ are N-dimensional unitaries dependant on the trainable parameters $\theta$ and $S^{(l)}$ are N-dimensional encoding unitaries of the form $S^{(l)} = \text{diag}(e^{-ix\lambda_l^1}, \ldots, e^{-ix\lambda_l^N})$ where $\lambda_l^i$s are eigenvalues of the underlying encoding Hamiltonian $H_l$ (without loss of generality).

We start by applying the first layer on the $|0\rangle$ computational basis state and we iterate through the remaining layers to obtain $|\psi(x; \theta)\rangle = U(x; \theta) |0\rangle$.

$$S^{(1)}(x)W^{(1)}\,|0\rangle = \sum_{j_1=1}^{N} W_{j_1 1}^{(1)} e^{-ix\lambda_1^{j_1}}\,|j_1\rangle$$

$$S^{(2)}(x)W^{(2)}S^{(1)}(x)W^{(1)}\,|0\rangle = \sum_{j_2=1}^{N}\sum_{j_1=1}^{N} W_{j_2 j_1}^{(2)} W_{j_1 1}^{(1)} e^{-ix(\lambda_1^{j_1}+\lambda_2^{j_2})}\,|j_2\rangle$$

$$\vdots$$

$$W^{(L+1)}\prod_{l=1}^{L} S^{(l)}(x)W^{(l)}\,|0\rangle = \sum_{k=1}^{N}\sum_{\boldsymbol{j}\in[N]^L} W_{kj_L}^{(L+1)}\ldots W_{j_1 1}^{(1)} e^{-ix\Lambda_{\boldsymbol{j}}}\,|k\rangle$$

The full quantum model will consequently have the form of a truncated Fourier series:

$$f(x;\theta) = \langle 0|U(x;\theta)^{\dagger} O U(x;\theta)|0\rangle$$

$$= \sum_{k'=1}^{N}\sum_{\boldsymbol{j'}\in[N]^L}\sum_{k=1}^{N}\sum_{\boldsymbol{j}\in[N]^L} (W_{k'j_L'}^{(L+1)}\ldots W_{j_1'1}^{(1)})\dagger\,\langle k'|O|k\rangle\,(W_{kj_L}^{(L+1)}\ldots W_{j_1 1}^{(1)}) e^{-ix(\Lambda_{\boldsymbol{j}}-\Lambda_{\boldsymbol{j'}})}$$

$$= \sum_{\boldsymbol{j},\boldsymbol{j'}\in[N]^L} c_{\boldsymbol{j},\boldsymbol{j'}} e^{-ix(\Lambda_{\boldsymbol{j}}-\Lambda_{\boldsymbol{j'}})}$$

$$= \sum_{\omega\in\Omega} c_{\omega} e^{-ix\omega}$$

In this proof, we considered encoding unitaries of size $N = 2^n$ with $n$ the number of qubits. However, we note that, without loss of generality, we can use encoding gates applied only on subsets of the n qubits and consider the eigenvalues of the global resulting hamiltonian which are no different than the ones of the local hamiltonian with some redundancies.

$$\square$$

The simplest case is called Pauli encoding, when all encoding Hamiltonians are in fact Pauli matrices (e.g. encoding gates $R_Z(x) = e^{-i\frac{x}{2}\sigma_Z}$) as in Schuld et al. (2021); Caro et al. (2021). In this case, all the eigenvalues are $\lambda = \pm 1/2$, and therefore, the $\Lambda_{\boldsymbol{i}}$ are all integers (or half-integers, if $L$ is odd) in $[-L/2, L/2]$. It follows that the set of distinct values in $\Omega$ is simply the set of integers in $[\![-L, L]\!]$. Indeed, in this case, there are many redundant frequencies, due to the fact that all Pauli eigenvalues are the same. As shown in Figure 2, various eigenvalues would create more distinct frequencies in the end. In the rest of the paper, $\Omega$ will denote the set of unique frequencies, without redundancy.

Note that in Section 4, we observe an unexpected phenomenon: it seems that redundant frequencies are likely to have high coefficients (for both random and trained VQCs). Unique frequencies might often have in contrast small coefficients, reducing the potential expressivity of the VQC. We see that the redundancy might therefore play an important role in the expressivity of VQCs, and leave theoretical proof for future work. In fact, the impact of the frequencies redundancy on the quantum model has been recently highlighted in Peters & Schuld (2022) where an analytical correspondence between a frequency redundancy and its Fourier coefficient has been proven for a simplified class of parameterized quantum models.

These arguments give some intuition on why one should use encoding gates from Hamiltonians with rich and various eigenvalues, by taking complex interactions over many qubits. A global Hamiltonian over $n$ qubits, hard to implement, could potentially have $2^n$ distinct eigenvalues, thus enlarging $\Omega$ and avoid redundancy. Another approach from Kyriienko et al. (2022); Shin et al. (2022) consists in adding scaling factors in the Pauli encoding gate to modify their eigenvalues and avoid redundancy. It results in an exponential number of integer frequencies, with respect to $L$, with many very high frequencies.

We can now generalize, if we now suppose that $\mathcal{X} = \mathbb{R}^d$, such that we encode a vector $x = (x_1, \ldots, x_d)$ in our quantum model, then $\Omega$ becomes the following $d-$dimensional Cartesian product $\Omega = \Omega_1 \times \Omega_2 \times \cdots \times \Omega_d$, where each $\Omega_\kappa$ is defined as in Eq.7 on its own set of Hamiltonians.

In this context, note that the frequencies $\omega$ are now vectors in $\mathbb{R}^d$ and there are $d$ different trees to build $\Omega$ (see Figure 2). Note that for notation simplicity, we assumed that $L$ gates were applied on each input's component, but it can be generalized to any number of gates per dimension.

We therefore see that the size of the spectrum $|\Omega|$ can potentially grow exponentially with the number of encoding gates and the dimension of the input data. For instance, consider a $d$-dimensional vector $x$ and $L$ Pauli-encoding gates for each dimension in such a way that there are $Ld$ encoding gates in the VQC. The size of the spectrum $\Omega$ would scale as $O(L^d)$, which becomes quickly intractable as $d$ increases.

As an example, the spectrum associated to a VQC with $L = 20$ encoding gates and $d = 16$ would require more than one hundred times the world's storage data capacity available in 2007 to be stored (Hilbert & López (2011)). We therefore wonder if it is possible to build a classical approximator $\tilde{f}(x) = \sum_{\omega \in \tilde{\Omega}} \tilde{c}_\omega e^{i\omega x}$, such that $\tilde{\Omega}$ is of tractable size and $\sup_{x \in \mathcal{X}} \left\| f(x) - \tilde{f}(x) \right\| \leq \varepsilon$.

### B.3 QUANTUM MODELS ARE SHIFT-INVARIANT KERNEL METHODS

As the quantum model is a real-valued function, it follows that $\omega \in \Omega$ implies $-\omega \in \Omega$ and $c_\omega = c_{-\omega}^*$. We express the Fourier series of the quantum model as a sum of trigonometric functions by defining for every $\omega \in \Omega$:

$$a_\omega := c_\omega + c_{-\omega} \in \mathbb{R} \tag{8}$$

$$b_\omega := \frac{1}{i}(c_\omega - c_{-\omega}) \in \mathbb{R} \tag{9}$$

such that

$$
\begin{aligned}
f(x; \theta) &= \sum_{\omega \in \Omega_+} c_\omega e^{i\omega x} + c_{-\omega} e^{-i\omega x} \\
&= \sum_{\omega \in \Omega_+} a_\omega \cos(\omega x) + b_\omega \sin(\omega x)
\end{aligned}
\tag{10}
$$

where $\Omega_+$ contains only half of the frequencies from $\Omega$. Considering only Pauli matrices, if $d = 1$, we simply have $\Omega = [\![-L, L]\!]$ and $\Omega_+ = [\![0, L]\!]$. In dimension $d$, we have $\Omega = [\![-L, L]\!]^d$ and $\Omega_+$ is built by keeping half of the frequencies (after removing those of opposite sign), plus the null vector. In the end, we have

$$|\Omega_+| = \frac{(2L + 1)^d - 1}{2} + 1 \tag{11}$$

With a more general encoding scheme, if there is a different number of distinct positive frequencies per dimension, the formula is different but is built similarly.

In the following parts, we will focus solely on $\Omega_+$ and conveniently drop the $+$ subscript.

Given this formulation of the quantum model, we define the feature map of the quantum model as

$$f(x; \theta) = \langle \psi(x; \theta) | O | \psi(x; \theta) \rangle = \mathbf{w}(\theta)^T \phi(x) \tag{12}$$

where $\phi(x)$ is the *feature vector*, the mapping of the initial input into a larger *feature space*, where the new distribution of the data is supposed to make the classification (or regression) solvable with only a linear model. This linear model is in fact the inner product between $\phi(x)$ and a trainable *weight vector* $\mathbf{w}$. In the case of VQCs, we can explicitly express them as:

$$\phi(x) = \frac{1}{\sqrt{|\Omega|}} \begin{bmatrix} cos(\omega^T x) \\ sin(\omega^T x) \\ \vdots \end{bmatrix}_{\omega \in \Omega}, \quad \mathbf{w}(\theta) = \begin{bmatrix} a_\omega \\ b_\omega \\ \vdots \end{bmatrix}_{\omega \in \Omega} \tag{13}$$

If the spectrum $\Omega$ is known and accessible, one can fit the quantum model by retrieving the co-efficients $a_\omega, b_\omega$ associated to each frequency $\omega$. This can be done by using general linear ridge regression techniques. Interestingly, there exists a dual formulation of the linear ridge regression that depends entirely on the kernel function associated to the model (Bishop & Nasrabadi (2006)). The related kernel function is defined by:

$$
\begin{aligned}
k(x, x') &= \langle \phi(x), \phi(x') \rangle \\
&= \frac{1}{|\Omega|} \sum_{\omega \in \Omega} \cos(\omega x)\cos(\omega x') + \sin(\omega x)\sin(\omega x') \\
&= \frac{1}{|\Omega|} \sum_{\omega \in \Omega} \cos(\omega(x - x'))
\end{aligned}
\tag{14}
$$

which is a shift-invariant kernel, meaning that $k(x, x') = k(x - x')$.

It is known that quantum models from VQCs are equivalent to kernel methods (Schuld (2021)), which means that it is equally possible to fit the quantum model by approximating the related kernel function. These kernels are high dimensional (since the frequencies in $\Omega$ can be numerous) which makes it hard to simulate classically in practice. But due to their shift-invariance, we propose to study their classical approximation using Random Fourier Features (RFF), a seminal method known to be powerful approximator of high-dimensional kernels (Rahimi & Recht (2009)).

## C  DEFINITIONS OF LINEAR RIDGE REGRESSION (LRR) AND KERNEL RIDGE REGRESSION (KRR)

We present in this Section the Linear Ridge Regression (LRR) and Kernel Ridge Regression (KRR) problem (Bishop & Nasrabadi (2006)). The problem of regression is to predict continuous label values from feature vectors. We are given a dataset $\{(x_i, y_i), i \in [\![1, M]\!] x_i \in \mathbb{R}^d, y_i \in \mathbb{R}\}$, and to each data point $x$ an associated feature vector $\phi(x) \in \mathbb{R}^p$. The goal of LRR is to construct a parameterized model $f$ such that $f(x) = y$. The model is parameterized by a weight vector $\mathbf{w}$ of size $p$ such that $f(x; w) = \mathbf{w}^T \phi(x)$. Training the model consists of finding the vector $\mathbf{w}*$ that minimizes the loss function

$$
\mathbf{w}^* = \arg\min_{\mathbf{w}} \frac{1}{M} \sum_{i=1}^{M} |\mathbf{w}^T \phi(x_i) - y_i|^2 + \lambda ||\mathbf{w}||^2
\tag{15}
$$

$$
= \arg\min_{\mathbf{w}} \frac{1}{M} ||\mathbf{\Phi}\mathbf{w} - \boldsymbol{y}||^2 + \lambda ||\mathbf{w}||^2
\tag{16}
$$

$$
\tag{17}
$$

where $\mathbf{\Phi}$ is a matrix of size $M \times p$ with each row $i$ corresponds to $\phi(x_i)^T$ and $\boldsymbol{y}$ is the vector of all the labels $y_i$. The first term of the loss is the Mean Square Error (MSE) and corresponds to the difference between the prediction and the ground truth. The second term is the ridge regularization, and prevents the weights from exploding. The magnitude of the regularization is controlled by the hyperparameter $\lambda > 0$.

When $p < M$, an analytic solution to this problem is given by $\mathbf{w}^* = (\mathbf{\Phi}^T\mathbf{\Phi} + M\lambda I_p)^{-1}\mathbf{\Phi}^T\boldsymbol{y}$.

As a consequence, to make the LRR possible and have a single solution, the number of training points must be larger than the number of features in the feature space ($\phi(x)$). Otherwise, one can perform a gradient descent.

The dual formulation of this problem is given by expressing $\mathbf{w}$ as a linear combination of the data points $\mathbf{w} = \mathbf{\Phi}^T \boldsymbol{\alpha}$. The minimization on $\mathbf{w}$ become a minimization on $\boldsymbol{\alpha}$ and can be expressed as

$$
\boldsymbol{\alpha}^* = \arg\min_{\boldsymbol{\alpha}} \frac{1}{M} ||\mathbf{\Phi}\mathbf{\Phi}^T\boldsymbol{\alpha} - \boldsymbol{y}||^2 + \lambda \boldsymbol{\alpha}^T \mathbf{\Phi}\mathbf{\Phi}^T \boldsymbol{\alpha}
\tag{18}
$$

$$
\tag{19}
$$

The solution of this problem is $\boldsymbol{\alpha} = (\mathbf{\Phi}\mathbf{\Phi}^T + M\lambda I_M)^{-1}\boldsymbol{y}$.

Note that the dual solution only depends on the matrix of scalar products between feature vectors $\mathbf{\Phi}\mathbf{\Phi}^T$. One can then replace this matrix by a kernel matrix $K$ and the obtained model is a Kernel Ridge Regression.

## D  APPROXIMATION RESULTS FOR RANDOM FOURIER FEATURES

We give here two useful results about the bounds of the error of the RFF method. RFFs are supposed to approximate a certain kernel $k$ by using fewer features. Intuitively, not enough features would lead to imprecise solutions. The following theorems (Rahimi & Recht (2009); Sutherland & Schneider (2015)) bounds the error obtained when comparing the kernel $k(x, y)$ by the RFF approximator $\phi(x)^T\phi(y)$ using $D$ samples.

We recall that the condition on the kernel $k$ is for it to be expressed as

$$k(\delta) = \int_{\omega \in \mathcal{X}} p(\omega)e^{-i\omega^T\delta}d\omega \tag{20}$$

where $p(\omega)$ is the distribution of the frequencies $\omega$.

**Theorem 2.** *Let $\mathcal{X}$ be a compact set of $\mathbb{R}^d$, and $\epsilon > 0$.*

$$\mathbb{P}(\sup_{x,y\in\mathcal{X}} |k(x - y) - \phi(x)^T\phi(y)| \geq \epsilon) \leq$$

$$66(\frac{\sigma_p|\mathcal{X}|}{\epsilon})^2 exp(-\frac{D\epsilon^2}{4(d+2)}) \tag{21}$$

*with $\sigma_p^2 = \mathbb{E}_p(\omega^T\omega)$, the variance of the frequencies' distribution, and $|\mathcal{X}| = max_{x,x'\in\mathcal{X}}(\|x-x'\|)$ the diameter of $\mathcal{X}$.*

The following theorem (Sutherland & Schneider (2015)) bounds the actual prediction error when using RFF compared to the KRR estimate. The formula in the original reference contains a sign error and we correct it here.

**Theorem 3.** *Let $\mathcal{X}$ be a compact set of $\mathbb{R}^d$, and $\epsilon > 0$. We consider a training set $\{(x_i, y_i)\}_{i=1}^M$. Let $f$ be the KRR model obtained with the true kernel $k$ and regularization $\lambda = M\lambda_0$ for $\lambda_0 > 0$, and $\tilde{f}$ the KRR model obtained with the approximate kernel and the same regularization. Then we can guarantee $|f(x) - \tilde{f}(x)| \leq \epsilon$ with probability $1 - \delta$ for a number $D$ of samples given by:*

$$D = \Omega\left(d\left(\frac{(\lambda_0 + 1)\sigma_y}{\lambda_0^2\epsilon}\right)^2\left[log(\sigma_p|\mathcal{X}|) + log\frac{(\lambda_0 + 1)\sigma_y}{\lambda_0^2\epsilon} - log\delta\right]\right) \tag{22}$$

*with $\sigma_y^2 = \frac{1}{M}\sum_{i=1}^M y_i^2$ and $\sigma_p$, $|\mathcal{X}|$ being defined in theorem 2. We recall that in Eq.22 the notation $\Omega$ stands for the computational complexity "Big-$\Omega$" notation.*

## E  APPROXIMATION RESULTS FOR RFF IN THE CONTEXT OF VQCS

### E.1  DISTINCT SAMPLING IN THE PAULI ENCODING CASE

In the case of Pauli encoding only, we know that $\Omega = [\![-L, L]\!]^d$ (considered here to be the full spectrum, not $\Omega_+$ defined in Section B.3, which would have been equivalent). In one dimension, we simply have $\sigma_p = 1/L\sum_{\ell=-L,\cdots,L}\ell^2 = O(L^2)$. In dimension $d$, a frequency $\omega$ is given by its values on each dimension $(j_1, \cdots, j_d)$ with $j_k \in [|-L, L|]$. We similarly have

$$\sigma_p = \frac{1}{(2L+1)^d}\sum_{j_1,\cdots,j_d} j_1^2 + \cdots + j_d^2 \tag{23}$$

Note that $\sum_{j_1,\cdots,j_d} j_1^2 + \cdots + j_d^2$ is $d(2L+1)^{d-1}$ times the sum of all squares,

$$\sigma_p = \frac{d(2L+1)^{d-1}}{(2L+1)^d}\sum_{\ell=-L}^L \ell^2 = \frac{d}{2L+1}\frac{2L(L+1)(2L+1)}{6} \tag{24}$$

$$= O(dL^2) = O(d|\Omega|^{2/d})$$

The expression is then obtained by replacing the value of $\sigma_p$ in theorem 3.

We note that we can generalize this results to scaled Pauli encoding, as done in Kyriienko et al. (2022); Shin et al. (2022), by replacing $L$ by a term growing as $c^L$ where $c$ is a constant. $D$ would grow linearly in $L$ and not logarithmically anymore.

## E.2 GRID SAMPLING WITH A GENERAL HAMILTONIAN

We provide here a bound on the minimum of samples required to achieve a certain error between the RFF model and the complete model in the case of a general encoding in the gird sampling strategy. The proof and details for this theorem is shown in Appendix Section E.2.

**Theorem 4.** *Let $\mathcal{X}$ be a compact set of $\mathbb{R}^d$, and $\epsilon > 0$. We consider a training set $\{(x_i, y_i)\}_{i=1}^{M}$. Let $f$ be a VQC model with any hamiltonian encoding, with a maximum individual frequency $\omega_{\max}$ and full freedom on the associated frequency coefficients, trained with a regularization $\lambda$. Let $\sigma_y^2 = \frac{1}{M}\sum_{i=1}^{M} y_i^2$ and $|\mathcal{X}|$ the diameter of $\mathcal{X}$. Let $\tilde{f}$ be the RFF model with $D$ samples in the grid strategy trained on the same dataset and the same regularization. Let $C = |f|_\infty |\mathcal{X}|$ and $s$ the sampling rate defined in the grid sampling strategy. Then we can guarantee $|f(x) - \tilde{f}(x)| \leq \epsilon$ for $0 < s < \frac{1}{C}$ with probability $1 - \delta$ for a number $D$ of samples given by:*

$$D = \Omega\left(\frac{dC_1}{\lambda^4(\epsilon - sC)^2}\left[log(\omega_{\max}|\mathcal{X}|) + log\frac{C_2}{\lambda^2(\epsilon - sC)} - log\delta\right]\right) \tag{25}$$

*with $C_1$ and $C_2$ being constants depending on $\sigma_y$, $d(X)$. We recall that in Eq.25 the notation $\Omega$ stands for the computational complexity "Big-$\Omega$" notation.*

*Proof.* The following theorem bounds the approximation between a function defined by its Fourier series and another function with frequencies distant by at most a constant $s$ of the original frequencies.

Let $\mathcal{X}$ a compact set of $\mathbb{R}^d$ with diameter $|\mathcal{X}|$ and $\Omega$ a finite subset of $\mathcal{X}$. Let $f(x) = \sum_{\omega \in \Omega} a_\omega cos(\omega^T x) + b_\omega sin(\omega^T x)$. Let $\Omega'$ a subset of $\mathcal{X}$ and $s > 0$ such that $\forall \omega \in \Omega$, $\exists \omega' \in \Omega$, $|\omega - \omega'| \leq s$.

Let $\mathcal{F}_{\Omega'} = \left\{\sum_{\omega \in \Omega'} a_\omega cos(\omega^T x) + b_\omega sin(\omega^T x), a_\omega, b\omega \in \mathbb{R}\right\}$.

**Theorem 5.** *It exists f' $\in \mathcal{F}_{\Omega'}$ such that*

$$\sup_{x \in \mathcal{X}} |f'(x) - f(x)| \leq sC \tag{26}$$

*with $C = |\mathcal{X}||f|_\infty$.*

*Proof.* For each $\omega \in \Omega$ let $b(\omega) \in \Omega'$ be such that $|\omega - b(\omega)| \leq s$. Such element exists by definition but is not necessarily unique. Let $f'(x) = \sum_{\omega \in \Omega} a_\omega cos(b(\omega)^T x) + b_\omega sin(b(\omega)^T x)$. The $b(\omega)$s are not necessarily different therefore there might be less frequencies in $f'$ than in $f$.

$$|f(x) - f'(x)| = 2\left|\sum_{\omega \in \Omega} sin(\frac{(\omega - b(\omega))^T}{2}x)\right. \tag{27}$$

$$\left.[b_\omega sin(\frac{(\omega + b(\omega))^T}{2}x) - a_\omega cos(\frac{(\omega + b(\omega))^T}{2}x)]\right| \tag{28}$$

$$\leq 2\sum_{\omega \in \Omega}|\frac{(\omega - b(\omega))^T}{2}||x|[|b_\omega| + |a_\omega|] \tag{29}$$

$$\leq s|x|\sum_{\omega \in \Omega}|b_\omega| + |a_\omega| \tag{30}$$

$$\leq s|\mathcal{X}||f|_\infty \tag{31}$$

$$\square$$

We shall here extend the proof where we sample from the grid described above. Let us note $\hat{f}_s$ the RFF model with the whole grid and $\tilde{f}$ the RFF model with D samples from the grid below. For all $x \in \mathcal{X}$ we have

$$|\tilde{f}(x) - f(x)| \leq |\tilde{f}(x) - \hat{f}_s| + |\hat{f}_s - f(x)| \tag{32}$$

$$\leq |\tilde{f}(x) - \hat{f}_s| + sC \tag{33}$$

Then

$$\mathbb{P}(|\tilde{f}(x) - f(x)| \geq \epsilon) \leq \mathbb{P}(|\tilde{f}(x) - \hat{f}_s| \geq \epsilon - sC) \tag{34}$$

for $s < \epsilon/C$.

In this case $|\Omega| = (\omega_{\max}/s)^d$ Using the expression of Section E.1, we can guarantee that $|f(x) - \tilde{f}(x)| \leq \epsilon$ with probability $1 - \delta$ if

$$D = \Omega\left(d\frac{1}{(\epsilon - sC)^2}\left[log(\omega_{\max}/s) + log\frac{1}{\epsilon - sC} - log\delta\right]\right) \tag{35}$$

$$\square$$

## F    LIMITATIONS OF RFF FOR APPROXIMATING VQCS

In Section 3, we have seen the theoretical power of Random Fourier Features and three different adaptations to approximate VQCs in practice. Since many parameters are to be taken into account (size and structure of $\Omega$, number of qubits, circuit depth, number of training points, input dimension, encoding Hamiltonians, etc.), it is natural to ask ourselves in which of the three strategy is recommended given a use case, and are there any use cases for which none of them work.

As seen in Section 3.4, we know the lower bound on the number of samples to draw in RFF, to reach a specific error. This bound grows linearly with the input dimension $d$, and logarithmically with the size of $\Omega$ (itself depending exponentially on $L^d$). Nonetheless, in practice, we could see very large spectrum to be harder to approximate, simply because it would require much more samples. This scaling will be judged once such VQCs will be actually implemented on large enough quantum computers (with enough qubits and/or long coherence).

$\Omega$ increases as well when the encoding Hamiltonians have distinct eigenvalues and are acting on many qubits. Therefore, quantum computers allowing for many qubits and various high locality Hamiltonians would be a plus for enlarging the spectrum.

As the Hamiltonians become larger we could reach a limit where it becomes impossible to diagonalize them. In such a case, without sampling access to $\Omega$, the *Distinct* and *Tree* sampling strategies would be unavailable. The *Grid* sampling scheme would suffice until suffering from the high dimensionality or other factors detailed above.

Finally, having a small dataset would limit the trainability of our classical RFF methods. Note that this would probably constrain the training of the VQC as well.

Overall, some limits for our classical methods can be guessed and observed already, but the main ones remain to be measured on real and larger scale quantum computers. We leave this research for future work. On another hand, one could want to understand better the relation between the available frequencies and their amplitude in practice, to find potential singularities that could help, or not, the VQCs.

Finally, we want to insist on the fact that the assumptions on VQCs are crucial on the whole construction that we propose, and that some of them could be questioned, especially concerning the encoding. For instance, when encoding vectors $x = (x_1, \cdots, x_d)$, not having encoding gates expressed as $exp(-x_i H)$ could potentially change the expression of $f(x; \theta)$ (Eq.5) and therefore could change the fact that the associated kernel would be easily expressed as a Fourier series, with shift-invariance. For instance, in Kyriienko et al. (2021), the authors use $exp(-arcsin(x_i)H)$ to encode

data, resulting in $f$ being expressed in the Chebyshev basis instead of the Fourier one. More generally, understanding what happens with encodings of the form $exp(-g(x_i)H)$, and whether we can still use our classical approximation methods, remain an open question. Similar questions arise if we use simultaneous components encoding $exp(-x_i x_j H)$, or other alternative schemes.

## G  NUMERICAL SIMULATIONS: ADDITIONAL DETAILS

### G.1  METHODS AND DEFINITIONS

As shown in Figure 6, a typical random VQC instance is built from a list of general encoding Hamiltonians $\{H_1, \cdots, H_k\}$, applied to randomly selected qubits according to their locality. The number of qubits is fixed to 5 in all the experiments (Note that the number of qubits has no impact on the expressivity a priori).

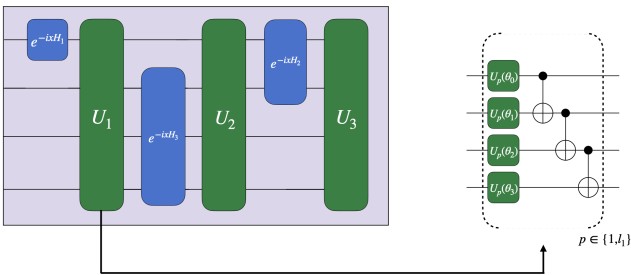

Figure 6: **Random instance of a VQC.** In this example, three encoding Hamiltonians $\{H_1, H_2, H_3\}$ are randomly assigned over four qubits, and load a 1-dimensional vector $x$. Following each encoding gate $H_i$, an ansatz with trainable parameters and a ladder of CNOTs is applied, $l_i$ times in a row.

### G.2  RFFs APPROXIMATION ON OTHER TYPES OF RANDOM VQCs

We introduce this Section by adding some details on the experimental framework described in Section 4.1. The random VQCs follow the structure shown in Figure 6. The training dataset we use is $\{X_{grid}, Y_{grid}\}$, with $X_{grid}$ being a set of $d$-dimensional data points spaced uniformly on the interval $\prod_{i=1}^{d}[0, x_{max_i}]$ and $Y_{grid}$ the evaluation of the quantum circuit on the input dataset $X_{grid}$.

We note that the number of data points in $X_{grid}$ needed to efficiently learn the quantum function is $N > \prod_{i=1}^{d} \frac{x_{max_i} w_{max_i}}{\pi}$. This choice is basically related to the *Shannon criterion* for effective sampling in order to reconstruct the full function covering all of its frequencies. Moreover, it is better for the solution to be unique and hence for the least square problem introduced in Eq.2 to be well defined, we choose $N$ to be bigger than the number of features in the regression problem (these two criteria coincide in the case of Pauli encoding).

### G.2.1  PAULI ENCODING

We first consider a quantum model with $L$ Pauli encoding gates per feature resulting in an integer-frequency spectrum (half of $[\![ -L : L ]\!]^d$). In this case, the corresponding quantum model is a periodic function of period $T = (2\pi)^d$ and thus, we choose $x_{max} = 2\pi$ for $X_{grid}$ construction.

In Figure 7, we implement VQCs with $L$=200 Pauli encoding gates, for a 1-dimensional input. We observe that our classical approximation methods are indeed able to reproduce such VQCs. On average, the RFF training error for *Distinct* and *Grid* sampling is a linear function of the number $D$ of samples taken from $\Omega$. On the other hand, the error using *Tree* sampling exhibits a faster decreasing trend, reaching relatively low errors with only 20% of the spectrum size. Indeed, the redundancy of Pauli encoding is extremely high, since with $L = 200$ gates, $\Omega$ can potentially have $3^{200}$ frequency, but only have 200 distinct ones, concentrated in the lower part.

We conjecture that the efficiency of *Tree* sampling is closely related to the redundancy in the discrete frequency distribution over $\Omega$. In fact, as shown in Figure 7, Fourier coefficients of the VQC are, on

average, correlated to the frequency redundancy in the empirical quantum spectrum. Frequencies above a certain threshold $\omega_{\text{effective}}$ are merely redundant for this particular encoding scheme, and we observe that they are cut from the quantum model empirical spectrum. The effective spectrum of the VQC is therefore smaller than what the theory predicts. Consequently, the fast decreasing trend of the *Tree* sampling stems from the fact that we sample accordingly to the redundancy, therefore requiring less frequency samples. We see that $0.2 \times |\Omega|$ samples are sufficient to sample approximately all frequencies in $[|0, \omega_{\text{effective}}|]$.

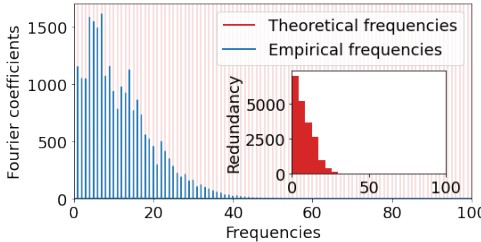 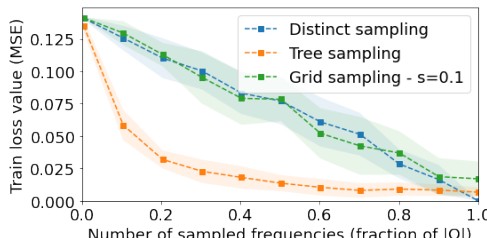

(a) Average Fourier Transform of the VQC's quantum models. The frequencies with high coefficients are the ones with high redundancy in $\Omega$ (seen in the inner red histogram). Frequencies over 100 have negligible coefficients and redundancy, and therefore are not shown.

(b) Evolution of RFF train loss as a function of the relative number of frequencies sampled. The *Tree* sampling strategy takes advantage of the high redundancy to sample less frequencies to reach a good approximation.

Figure 7: Random 1d VQCs with L=200 Pauli encoding gates, averaged over 10 different random initialization.

In Figure 8, we show a similar simulations with a $d$-dimensional input ($d = 4$) and $L = 5$ Pauli gates per dimension. According to Eq.(11), the theoretical number of distinct positive frequencies is 7321. In this case in the tree sampling procedure, we can sample both a frequency and its opposite without removing one of them. Therefore the scheme is a bit less performant than in dimension 1.

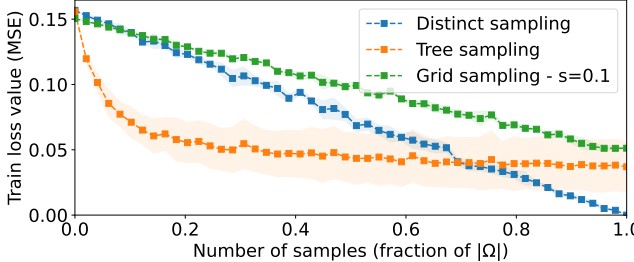

Figure 8: RFF performance for $L = 5, d = 4$, to approximate random VQCs with Pauli encoding.

### G.2.2 MORE COMPLEX HAMILTONIAN ENCODINGS

For Pauli encoding, we have seen that *Tree* sampling is highly effective for approximating the quantum model. Consequently, we designed VQCs with different spectrum distributions to study the RFF approximation performance in these cases.

As explained in Section 1, we consider encoding gates of the form $exp(-ix_i H)$ for each dimension $i$. One way to alter the spectrum distribution is the use of more general Hamiltonians $H$. To obtain exotic Hamiltonians while maintaining their physical feasibility (involving only two-bodies interactions), we use the generic expression

$$H_{XYZ} = \sum_{\langle i,j \rangle} \alpha_{ij} X_i X_j + \beta_{ij} Y_i Y_j + \gamma_{ij} Z_i Z_j + \sum_i \delta_i P_i \tag{36}$$

with the first term describing the interactions: $\langle i, j \rangle$ indicates a pair of connected particles and the second term describing a single particle's energy ($P_i = \{X_i, Y_i \text{ or } Z_i\}$).

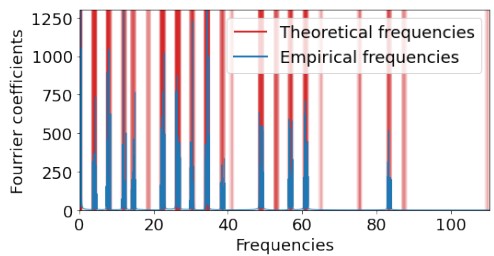 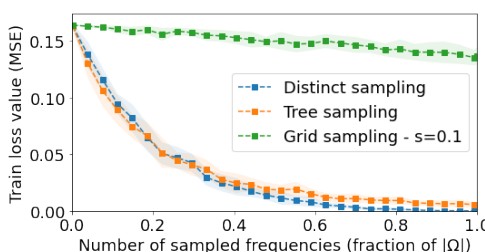

(a) Fourier transform averaged over different random initialization of the VQC. The intensity of the vertical red lines indicates the concentration of the theoretical frequencies in $\Omega$.

(b) RFF train loss with different sampling methods on the random VQCs. The Distinct sampling benefit from the concentration of frequencies in packets to approximate with less samples.

Figure 9: Random 1d VQCs with 4 scaled Paulis and and a 3-qubits $H_{XYZ}$ Hamiltonian

In Figure 9, we construct VQCs mixing both such Hamiltonians[1] and scaled Pauli[2] as encoding gates, on 1-dimensional inputs. In these cases, the corresponding quantum model is no longer $2\pi$-periodic, thus we have to find empirically a good value for $x_{max}$ (by increasing it until the performance reaches a limit).

With such complex encoding, we witness a different behavior for the *Distinct* sampling method, in comparison to the previous basic Pauli encoding scenario. Essentially, *Distinct* sampling has a faster than linear scaling, showing a clear and unexpected efficiency of RFFs in this case. We also notice that the *Tree* sampling method as a similar scaling. This observation points to the fact that, with the chosen encoding strategies, the frequencies in the spectrum $\Omega$ are concentrated in many packets or groups. This behavior is displayed with the concentrated red lines in Figure 9 (545 distinct frequencies out of 8192). Therefore, sampling just one of the many frequencies in a narrow packet is enough for the RFF to approximate it all. To put it differently, we can consider that there is a $\Omega_{\text{effective}}$ where each packet can be replaced by its main frequency, and the RFF manage to approximate it with fewer samples than the actual size of $\Omega$. To conclude, many distinct frequencies is not a guarantee of high expressivity.

As for *Grid* sampling, the choice of $s$ seemed too high for this solution to work in this case, in line with the theoretical bounds for this sampling method given in Theorem 1.

### G.2.3 EXPONENTIAL PAULI ENCODING

In order to obtain VQCs with a large number of frequencies, but low redundancy and no concentrated packets, we exploit the exponential encoding scheme proposed in Shin et al. (2022), resulting in a non degenerate quantum spectrum with $3^{Ld}$ distinct frequencies which is the maximal number of distinct frequencies we can get with two eigenvalues of opposing sign for each encoding Hamiltonian and thus an approximately uniform probability distribution over integers. In this encoding strategy, encoding Pauli gates are enhanced with a scaling coefficient $\beta_{kl}$ for the $l^{th}$ Pauli rotation gate encoding the component $x_k$. This gives us a total of $3^{Ld}$ positive and negative frequencies. These frequencies can be all distinct with the particular choice of $\beta_{kl} = 3^{l-1}$, resulting in an exponentially large and uniform $\Omega$. Note however that $\Omega$ is analytically known and contain only integer frequency, mostly very high frequencies for which the usefulness in practice remain to be studied.

We have tested our classical RFF approximation, shown in Figure 10, and obtain again the confirmation that RFF can approximate such an exponential feature space with a fraction of $|\Omega|$. This fraction might however be too large in practice. We also observe as expected that all three strategies have a linear scaling, in line with the absence of redundancy and frequency packets.

---

[1]in Figure 9, we used a 3-qubits Hamiltonian defined by: $H_{XYZ} = 7X_0X_1 + 7X_1X_0 + 0.11X_0X_2 + 0.1X_2X_0 + 8[Y_1Y_2 + Y_2Y_1 + Z_0Z_2 + Z_2Z_0]$

[2]Scaling factors are $[26.4309, 34.4309, 22.4309, 0.4309]$

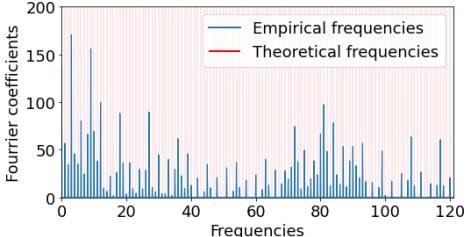 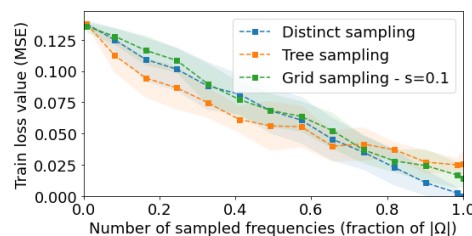

(a) Fourier transform averaged over different random initializations of the exponential encoding VQC with $L = 5$.

(b) RFF approximation performance of an the exponential encoding VQC with $L = 5$.

Figure 10: Random VQCs with exponentially large Spectrum, using $L = 5$ scaled Pauli encoding as in Shin et al. (2022) resulting in $\omega_{max} = 121$.

### G.3 ARTIFICIAL TARGET FUNCTION

We add here the training curve obtained during the training of the VQC with $L = 200$ Pauli encoding gates, on the artificial function $s(x) = \sum_{\omega \in \{4,10,60\}} cos(\omega x) + sin(\omega x)$. Despite the potential large number of frequencies available in $\Omega$, we have observed that the effective maximal frequency of the VQC was lower than 60, making it impossible for it to fit the high frequency of the target function.

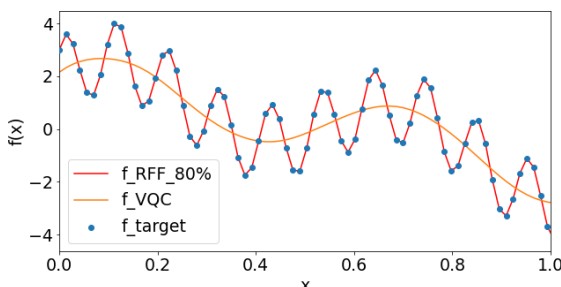

Figure 11: Predictions of the target function $s(x) = \sum_{\omega \in \{4,10,60\}} cos(\omega x) + sin(\omega x)$ with the quantum model and its corresponding RFF classical approximator using *Distinct* sampling.

### G.4 REAL DATASETS

For the binary classification task, we used the *PyTorch* Fashion MNIST dataset with the classes coat and dress (3 and 4). We divided the 12000 input data points into train and test datasets with $N_{train} = 9600$ and $N_{test} = 2400$. For the pre-processing, we downscaled the input dataset by first rescaling the flattened input images between 0 and 1 and subtracting the mean then performing a $d = 5$ PCA transformation fitted on the train data and applied on the test data. Finally, the 5-dimensional input vectors are rescaled between $-\pi$ and $\pi$. The final VQC predictions are obtained after 60 epochs using Adam optimizer with learning rate $= 0.01$ . For Tree sampling RFF training, for each fixed number of sampled frequencies $p$, we perform the regression on the corresponding fourier features using a *PyTorch* logistic regression model (linear layer $+$ sigmoid layer, loss : binary cross entropy with logits, metric : accuracy) trained for 2000 epochs with early stopping using Adam optimizer with learning rate $= 0.05$. The final accuracy for the fixed number of samples $p$ is the average score over 10 different such trained models with different random seeds for frequency sampling.

For the regression task, we used the *Scikit-learn* California housing dataset and kept only the first 5 features. We chose $N_{train} = 5000$ and $N_{test} = 1000$ and we scaled the 5-dimensional dataset between $-\pi$ and $\pi$. The final VQC predictions are obtained after 100 epochs using Adam optimizer with learning rate $= 0.01$. For Tree sampling RFF training, the same steps as in the case of the Fash-

ion MNIST dataset are performed with a *PyTorch* regression model (linear layer, loss and metric: mean squared error).

### G.5 Number of Samples and Size of $\Omega$

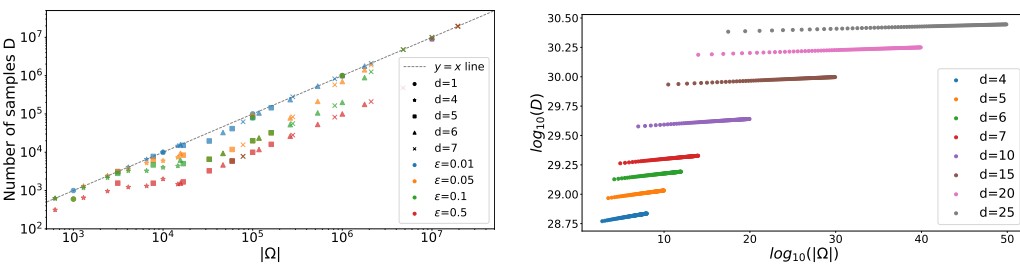

(a) Experimental bounds for different values of $L, d$.    (b) Theoretical bound for different values of $L, d$.

Figure 12: **Evolution of D as a function of input dimension d and of L encoding gates per dimension, and theoretical bounds.** In agreement with the theoretical bound, the number of samples $D$ given as a fraction of $|\Omega|$ decreases with the growth of the data input dimension and the number of encoding gates.

In this Section, we test the theoretical bound provided by the theorem 4. Given a spectrum $\Omega = [\![0, L]\!]^d$, a Fourier series model trained on a specific dataset, the theorem bounds the necessary number of samples for a RFF model to approximate the original model with an $\epsilon$ error. This is an approximation to the Pauli encoding VQCs where the spectrum is $\Omega = [\![-L, L]\!]^d$. For fixed values of $L, d$, and a spectrum $\Omega = [\![0, L]\!]^d$, we implement the following protocol to test this bound:

- Generate a dataset of $10^5$ points sampled uniformly from $[0, 1]^d$ and a labels coming from a Fourier series on $\Omega$ with coefficients chosen uniformly from $[0, 1/\sqrt{|\Omega|}]$. Split in a train set and a test set with respective fractions .9 and .1.
- For each value of D in $\{1, k|\Omega|/10 \text{ for } k \in [\![1, 10]\!]\}$, sample $D$ frequencies from $\Omega$ without replacement, and train a linear ridge regression with $\lambda = 10^{-6}$ on the train set. We performed the training with a Adam optimizer, a learning rate of .001, and between 50 and 200 epochs depending on teh size of the dataset. Compute the output on the test set.
- Compute the mean absolute error between the output of the trained model with all the frequencies and the output of all other model. Select the model with the lowest number of samples that has an error below $\epsilon$.

The results of the application of this protocol are shown Figure 12. For the values of $|\Omega|$ between $10^4$ and $10^6$, one can see a significant reduction to the number of samples needed to approximate the whole model. For $\epsilon = .05$, one can expect to need only half of the spectrum, whereas for $\epsilon = .5$, one only need about 10% of the spectrum. The trend does not continue above $|\Omega| = 10^7$.

There are several limitations to this experiment. The main one is the limited training of the models. For the biggest values of $|\Omega|$ we limit ourselves to 50 epochs, which may be not enough to reach the optimal parameters, and thus blur the interpretation. Furthermore although the theorem is valid for every number of data points, the overparameterized regime where there are much more parameters than data points is known to exhibit unusual effects in linear regression (Hastie et al., 2022).

Given the choices of $\lambda$ and $\epsilon$, the theoretical bounds are very high for the regimes we experimentally tested, so they are not relevant. The effect that is quantified by the theory appears from $|\Omega| = 10^{30}$, e.g one need approximately $D = 10^{30}$ samples to approximate a $10^{40}$ frequency spectrum. That is still unfeasible on classical computer, so only standard benchmarks will state on the usefulness of the RFF methods.

