# OpenReview forum: "Classically Approximating Variational Quantum Machine Learning with Random Fourier Features"
_ICLR.cc/2023/Conference — ICLR 2023 poster_

### Official Review · Reviewer_WzGu · 2022-10-13

**Confidence:** 4
**Correctness:** 3
**Technical Novelty And Significance:** 2
**Empirical Novelty And Significance:** 3
**Recommendation:** 6

**Clarity, Quality, Novelty And Reproducibility:**

From my perspective, the clarity, quality, and novelty of this work are okay but still have space to improve -- see my comments above. The reproducibility of this work is great – an anonymous open-source package is provided.

**Strength And Weaknesses:**

From my perspective, the topic of simulating quantum computing by classical computing is in general interesting. In particular, considering that quantum computing devices will stay at the status of NISQ for a while, research about whether they can be simulated by classical devices are in general interesting. This submission to ICLR 2023 falls into this category, and it provides both theoretical and experimental results. In addition, the codes are open-source with an anonymous package provided.

Nevertheless, after reading through the paper, I have concerns from a few aspects:

1. First and foremost, I’m not sure if this paper provides enough evidence for simulating VQCs on classical computers efficiently in general. In particular, I’m worried about both Algorithm 1 and Algorithm 2, RFF with distinct sampling and RFF with tree sampling. They both diagonalize the Hamiltonians of the VQC’s encoding gates, but I don’t think it is doable in general – the Hamiltonian matrix scales exponentially with the number of qubits, i.e., a VQC on 50 qubits will in general have Hamiltonians of dimension 2^50 * 2^50. Diagonalization of such matrices is simply intractable on classical computers. In addition, the algorithms run LRR or KRR, but regression can also be slow in general. In all, I feel that the authors should give significantly more explanation about why their proposed algorithms are actually efficient on classical computers. In theory, Theorem 1 in Section 3.4 is a sample complexity result and does not investigate time complexity. In practice, the experiments are only conducted for Hamiltonians with 3 qubits or so. Either more evidence from theory or practice that the proposed algorithms have good time cost will be very helpful.

2. The technical novelty of this work needs to be better explained. On the one hand, this paper follows the paper Schuld (2021) heavily, which had proposed the idea of applying kernel methods to machine learning problems. The authors may want to further explain how this work departs from Schuld (2021). It seems that using random Fourier features is a main consideration different from Schuld (2021), but this goes to my concern on the other hand: it omits very important results on quantum-inspired classical algorithms based on random Fourier features, namely Yamasaki et al. https://proceedings.neurips.cc/paper/2020/hash/9ddb9dd5d8aee9a76bf217a2a3c54833-Abstract.html at NeurIPS 2020 and Yamasaki and Sonoda https://arxiv.org/pdf/2106.09028.pdf. These two papers have studied the ideas of using classical sampling-based methods under random features to solving quantum machine learning problems. A very detailed comparison is very necessary from my perspective.

3. Minor suggestions and typos in the paper:

- Introduction:
- - At many places, the references shall be listed with brackets, for instance in Page 1, Abbas et al. (2021); Cong et al. (2019); Huang et al. (2021) -> (Abbas et al. 2021; Cong et al. 2019; Huang et al. 2021). This should be applied in general for references as descriptions and not being the subject/object in a sentence.

- - There’s a sentence: “When it comes to trainability, it has been shown that these exponential spaces are in fact drawbacks McClean et al. (2018)”. This is the famous barren plateau paper. I personally found this a bit inappropriate because the main of having barren plateau is not having exponential-size space, but more like the stochasticity in many ansatzes for parametrized quantum circuits has variance decrease exponentially in the number of qubits. I guess the reasoning here needs to be more careful.

- Page 2, bottom: The other gates are called trainable -> Other gates are trainable.
figure 2 -> Figure 2 (and many other similar places)

- Page 3, Eq.11 -> Eq. (11)

- Page 3, A scaled version of Pauli encoding exist, where the Pauli are -> A scaled version of the encoding exists, where the Paulis are

- Page 6: It would be helpful to explain the intuition of Theorem 1, especially how this is proved.

- Page 13: How do we get Eq. (6) and (7), i.e., the eigenvalues have those forms? More discussions about the linear algebra behind will be preferred.

- Page 18: There is a broken link at Appendix F. Also a few other typos: section 3.4 -> Section 3.4, (Eq.5) -> Eq. (5).

**Summary Of The Paper:**

Variational quantum circuits constitute a main class of quantum algorithm on current noisy, intermediate scale quantum devices (NISQ). This paper studies approximating the result of VQCs by classical algorithms. Specifically, for VQCs with Hamiltonian encoding, this paper demonstrates how to use random Fourier features (RFF) to sample a few frequencies, build an equivalent low-dimensional kernel, and then solve by linear ridge regression (LRR) and kernel ridge regression (KRR).

**Summary Of The Review:**

In summary, I think the topic of this paper on classical simulation of near-term quantum computing is interesting, and the authors are also able to provide some theoretical and experimental results to justify their result. Nevertheless, from my perspective, there is notable issue with the efficiency of the proposed classical algorithm, technical contribution compared to prior arts, and the paper has space to improve in overall presentations and writings.

---

> ### Author Response · Authors · 2022-11-18
> **Reply to the Reviewer (2/2)**
>
> *(this is the second part of our replies)*
>
> - 2.a - “What is the difference with Schuld (2021) [...]”.
>
> Our works builds upon Schuld (2021) and answers some questions left opened, as written in the Related Work section (Appendix A). The novelty of this is paper is in several folds. **Theory**: Schuld’s work used the kernel method formalism to study the VQCs. In our work, we provide three practical classical algorithms using the RFF methods to mimic VQCs, and prove in details the implications. **Experiments**: We have tested our methods, which involves analyzing VQCs spectrum in depth, and tested them with more complex Hamiltonians, on more qubits than previous works. Our findings have also shed a light on the potential correlation between frequency’s redundancy and their coefficients, which could paved the way for a better understanding of VQC’s expressivity.
>
>
> - 2.b - “What is the difference with quantum-inspired classical algorithms based on random Fourier features, namely Yamasaki et al. at NeurIPS 2020 and Yamasaki and Sonoda https://arxiv.org/pdf/2106.09028.pdf. These two papers have studied the ideas of using classical sampling-based methods under random features to solving quantum machine learning problems [...]”
>
> This first reference is a proposal of a quantum algorithm for solving a learning problem with random features. The idea is that random fourier features can be improved by sampling a data dependent distribution called optimized distribution. In practice this optimized distribution can be intractable and the authors propose a quantum algorithm to speed up the sampling. The quantum algorithm they proposed is also not parameterized. The paper investigates the intersection of the same concepts as we do, ie random features sampling and quantum algorithms, but with a different goal and a different way. In our case, we investigate parameterized quantum learning models as Fourier series and devise classical approximation schemes of the quantum algorithm. In their case, they accelerate classical RFF using a quantum algorithm. The second reference by the same authors is an extension of the first work with more proofs and error bounds.
> We have included these references in our revised version for completeness, but despite some common language, our work is substantially different from them.
>
> - 3 -a “Barren plateaus is not due to exponential space size but stochasticity in VQC and exponentially decreasing variance [...]”
>
> Barren plateaus is indeed mentioned in the introduction, but we recall that it is not the topic of this work, which focuses solely on expressivity of VQCs (the link between the two is of course very important). Our mention of Barren plateaus is therefore very brief on not detailed. We have removed this sentence to avoid any misunderstanding. That being said, several recent results [arxiv:2105.14377 ,arxiv:2208.14057] showed that in fact the exponential space size of the parametrized unitary manifold is explicitly related to the exponential decreasing variance, and therefore to the presence of barren plateaus. We have added these citations to the introduction for completeness.
>
> - 3-b “Page 13: How do we get Eq. (6) and (7), i.e., the eigenvalues have those forms? More discussions about the linear algebra behind will be preferred.”
>
> Eq.(6) and (7) are both taken from the work of Schuld(2021). This result is used as a prior in many works, therefore we didn’t consider to write the proof at first. However, for clarity, and for the paper to be self-contained, we have added the detailed proof in the Appendix B.2 as suggested by the reviewer.
>
> All other minors remarks were taken into account in the revised version, and we thank the reviewer for this helpful list.

---

> > ### Comment · Reviewer_WzGu · 2022-11-26
> > **Official Reply**
> >
> > Thanks for the discussions and improvements on the paper. I'm happy to slightly increase my score to borderline accept.

---

> ### Author Response · Authors · 2022-11-18
> **Reply to the Reviewer (1/2)**
>
> We thank the reviewer for the insightful comments and suggestions for improvement. All remarks have been taken into account in the revised version. In addition, we provide here some answers that we think can help improve the clarity of the main contributions of our work. These points have also been made more explicit in the revised version.
>
> - 1.a ”I don’t think it is doable in general – the Hamiltonian matrix scales exponentially with the number of qubits”, “In practice, the experiments are only conducted for Hamiltonians with 3 qubits or so”
>
> The reviewer is correct in pointing out the intractability of diagonalizing global Hamiltonians on a large number of qubits. This point is emphasized in our manuscript (Section 3.3 and Appendix F): global (or very large) Hamiltonians would indeed prevent distinct and tree sampling from being applicable and is the reason for introducing grid sampling. However historically, platforms based on digital quantum computing decompose quantum circuits in (1,2)-qubit gates, and many proposed variational approaches are restricted to this architecture, making our distinct and tree sampling approach relevant. Applying global Hamiltonians can be done on quantum simulators such as neutral-atom quantum computers and our manuscript gives them a new argument towards advantage.
>
> - 1.b -  “The algorithms run LRR or KRR, but regression can also be slow in general [...]”.
>
> The reviewer rightfully points out that LRR and KRR suffer from caveats, in particular with respect to the size of the kernel used. However, since RFF based methods are not performing these classical algorithms on the whole kernel, but on a lower dimensional one (with D samples instead of the whole spectrum) these disadvantages don’t appear unless D becomes as large as the kernel size. We are therefore not in the “general” case here. For instance, regression has been proven efficient in the Random Fourier Feature literature even for infinite dimensional kernels. This discussion remains very interesting for this work, and we have added it in the revised version (Section 2).
>
> - 1.c ”Theorem 1 in Section 3.4 is a sample complexity result and does not investigate time complexity [...]”.
>
> The advice of the reviewer to add more details on the time complexity is relevant, and there is a direct link between the number of samples (D) and the time complexity, which is already given in Section 2. Namely, LRR time complexity is O(MD^2+D^3), which is efficient in the context of RFF since D is small compared to M (number of training points). The time complexity of gradient descent is O(DT) where T is the number of iterations. on the other hand, KRR is O(M^3) which is prohibitive, and not used in our case. But as we see D is an important parameter that tells if RFF is efficient, explaining why, as in the seminal work of RFF, the focus is made on the number of samples needed to approximate the quantum model.
>
> *(second part of the replies below)*

---

### Official Review · Reviewer_neuE · 2022-10-25

**Confidence:** 2
**Correctness:** 4
**Technical Novelty And Significance:** 3
**Empirical Novelty And Significance:** 4
**Recommendation:** 8

**Clarity, Quality, Novelty And Reproducibility:**

Clarity: The paper, method, and results are clearly presented. The only issue is some lack of clarity in the last two experiments, where it is not clear how the RFF are used and what did we learn from them.

Quality: While I am not an expert in quantum accelerators, as far as I can tell the methods and theoretical claims appear to be correct.

Novelty: The paper is very novel, and could have a significant impact on quantum ML approaches.

Reproducibility: Code for reproducing the results was provided, though I did not look into it or try to run it. Theoretical results appear to be correct.

**Strength And Weaknesses:**

Strengths:
* Clearly introducing the main concepts of VQC for ML, as well as RFF, making the paper accessible to a broad audience.
* Very convincing arguments for the limitations of current VQC approaches, while outlining where future effort should be focused. Quantum advantage is a topic of high interest, and specifically quantum advantage for ML. Any work that helps shed light on when quantum-based approaches can and cannot help is of great significance.

Weaknesses:
* From a practical perspective, it is difficult to tell (for a non-expert in the field) how representative the experiments are. Could it be that there are factors of scale that are limited due to availability of current quantum accelerators that might change the empirical results -- not unlike the case of neural networks showing their true potential once sufficient data and compute became available? How dependent is the analysis and empirical results on current hardware and the regime tested?
* The last two experiments are a bit confusing. Are you fitting the RFF to match the training data or to approximate the given VQC? If the former, then what do we learn from these experiments as opposed to any other classical ML technique? If the latter, then how come the test loss is better for RFF when it is trained to match the VQC?

**Summary Of The Paper:**

The paper questions the benefits of variational quantum circuits (VQC) for machine learning by demonstrating simple classical algorithms for approximating them. Specifically, VQC can be thought of as a multidimensional fourier series and so makes it a natural target for approximation via Random Fourier Features (RFF). The paper details three sampling strategies to sample frequencies for constructing random features to approximate a given VQC, as well as providing theoretical bounds on the approximation error. Under some assumptions RFF can efficiently approximate VQC as commonly implemented. The methods are tested on both synthetic and real datasets to demonstrate their effectiveness. The theoretical results also leave open the possibility of how VQC could still be beneficial for machine learning.

**Summary Of The Review:**

Based on the importance of finding when or if quantum accelerators can achieve quantum advantage for ML, I believe this paper should be accepted. The paper clearly lays out the logic and theoretical analysis for attacking the current methods for quantum acceleration of ML, while also pointing the path forward. It is only my lack of expertise in this field that makes me a tad hesitant and thus my low confidence score.

---

> ### Author Response · Authors · 2022-11-18
> **Reply to the Reviewer**
>
> We thank the reviewer for the positive feedback. The reviewer has clearly understood the main contributions of the paper, and is rightfully questioning some aspects of the experimental section:
>
> - "Could it be that there are factors of scale that are limited due to the availability of current quantum accelerators that might change the empirical results [...]”
>
> In fact, the experiments in the paper were not performed on quantum computers but on classical simulators of quantum computers. We have made this even more explicit in the revised version. That being said, theoretically, there is no difference between training a VQC on a quantum computer and on a classical simulator, apart from speed and limited scaling on the classical simulator. However, current quantum hardware available are too small and noisy to be of interest to our case. In conclusion, we don’t expect future quantum computers to change the behavior of theoretical findings. However, in the long term, with large enough quantum computers, we could enter a regime where, despite the good scaling of our classical sampling method, the VQC maintains a strong advantage over it in practice, as pointed out in the paper.
>
> - "The last two experiments are a bit confusing. Are you fitting the RFF to match the training data or to approximate the given VQC? [...]"
>
> In the last two simulations, we are indeed comparing the training of RFF and VQC on the same dataset. The reviewer is right, we could have also compared the VQC against any other ML method, but here the goal of this paper is to provide an equivalent model to the quantum circuit finds, with scalability guarantees. Focusing on the classical equivalent to the VQC is also a proper way to perform a fair comparison between quantum and classical. We have made this more explicit in the revised version.

---

### Official Review · Reviewer_tPCQ · 2022-10-25

**Confidence:** 3
**Correctness:** 4
**Technical Novelty And Significance:** 4
**Empirical Novelty And Significance:** 4
**Recommendation:** 8

**Clarity, Quality, Novelty And Reproducibility:**

The paper presents clear results showing the proposed sampling strategies can approximate VQCs efficiently. The work is novel and code for reproducing the results are included.

**Details Of Ethics Concerns:**

No ethics concerns

**Strength And Weaknesses:**

Very good theoretical derivations and experimental results. The results are well presented and extensive, covering both artificial and real datasets.

**Summary Of The Paper:**

This paper provides a classical method of approximating VQCs by sampling random frequencies from the VQC as a Fourier series. By showing that VQCs can be efficiently approximated using less than the order of exponential samplings, the work points out the potential problems with quantum advantages of VQCs.

**Summary Of The Review:**

A good study on approximating VQCs is presented with clear numerical evidence. The results are very interesting, pointing to potential flaws of VQCs which should be further studied in future works.

---

> ### Author Response · Authors · 2022-11-18
> **Reply to the Reviewer**
>
> We thank the reviewer for this positive assessment.

---

### Decision · Program_Chairs · 2023-01-20

**Decision:**

Accept: poster

**Justification For Why Not Higher Score:**

Topic might be of limited interest at ICLR at this point. A poster might attract attendees that have focused interest in this topic

**Justification For Why Not Lower Score:**

The reviewers are happy with the reported results and the theoretical justification of the methodologies proposed.

**Metareview: Summary, Strengths And Weaknesses:**

- Summary:

The paper questions the benefits of variational quantum circuits (VQC) for machine learning by demonstrating simple classical algorithms for approximating them. Specifically, VQC can be thought of as a multidimensional fourier series and so makes it a natural target for approximation via Random Fourier Features (RFF). The paper details three sampling strategies to sample frequencies for constructing random features to approximate a given VQC, as well as providing theoretical bounds on the approximation error. Under some assumptions RFF can efficiently approximate VQC as commonly implemented. The methods are tested on both synthetic and real datasets to demonstrate their effectiveness. The theoretical results also leave open the possibility of how VQC could still be beneficial for machine learning.

- Strengths
Findings of the paper include:
1. Very good theoretical derivations and experimental results.
2. Clearly introducing the main concepts of VQC for ML, as well as RFF, making the paper accessible to a broad audience.
3. The topic of this paper on classical simulation of near-term quantum computing is interesting

- Weaknesses
1. From a practical perspective, it is difficult to tell how representative the experiments are; some reviewers found the experiments a bit confusing.

- What would be missing:
1. Better clarity in parts of the paper, as suggested by reviewers


**Note From Pc:**

if the above contains the word "oral" or "spotlight" please see: "oral" presentation means -> notable-top-5% and "spotlight" means -> notable-top-25%. As stated in our emails, we are disassociating presentation type from AC recommendations